# Importance Weighted Score Matching for Diffusion Samplers with Enhanced Mode Coverage

## Abstract

Training neural samplers from unnormalized densities without target samples is challenging, particularly for achieving comprehensive mode coverage. In data-free scenarios, a fundamental discrepancy arises: while the training objective requires expectations over the target distribution, we can only sample from the model-induced distribution. Previous methods ignore this mismatch, resulting in mode-seeking objectives similar to reverse KL divergence. While recent approaches like replay buffers provide heuristic mitigation, they lack a principled correction for this distribution mismatch. In this work, we propose *Importance Weighted Score Matching*, a principled training approach for diffusion-based samplers that optimizes a mode-covering objective analogous to the forward KL divergence by re-weighting the score matching loss with tractable importance sampling estimates, thereby overcoming the absence of target distribution data. We also provide theoretical analysis of the bias and variance for the proposed estimator and the practical loss function used in our method. Experiments on increasingly complex multi-modal distributions, including 2D Gaussian Mixture Models with up to 120 modes and challenging particle systems with inherent symmetries, demonstrating that our approach consistently outperforms existing neural samplers across all distributional distance metrics, achieving state-of-the-art results on all benchmarks.

## 1 Introduction

Sampling from complex probability distributions is a fundamental challenge underpinning progress across diverse scientific disciplines, including physics (Metropolis et al., 1953; Noé et al., 2019; Albergo et al., 2019; Wirnsberger et al., 2020; Nicoli et al., 2020), chemistry (Frenkel & Smit, 2023; Noé et al., 2020), structural biology (Jumper et al., 2021; Bose et al., 2024), and machine learning tasks such as Bayesian inference (Gelfand & Smith, 1990; Box & Tiao, 2011). A particularly demanding scenario involves target distributions specified only via unnormalized density functions $\pi(x) \propto \exp(-E(x))$, where the normalizing constant remains intractable. Traditional Markov Chain Monte Carlo (MCMC) methods (Owen, 2013a; Metropolis et al., 1953; Neal et al., 2011; Del Moral et al., 2006), while foundational, face significant challenges in high-dimensional or multi-modal settings: extensive tuning requirements (Hoffman et al., 2014), slow mixing between separated modes (Neal, 2001; Neal et al., 2011), and high computational costs from sequential sample generation (Ren & Orkoulas, 2007), all hindering efficient acquisition of independent samples (Martino et al., 2015).

These limitations have motivated the development of deep neural networks as amortized samplers ("Neural Samplers") (Rezende & Mohamed, 2015; Dinh et al., 2017; Noé et al., 2019; Midgley et al., 2023b; Zhang & Chen, 2022; Berner et al., 2022; Vargas et al., 2023a; Akhound-Sadegh et al., 2024; He et al., 2024). A particularly challenging yet critical scenario arises when these samplers must be trained in a *data-free* setting (He et al., 2024), relying solely on the unnormalized density $E(x)$ without access to samples from $\pi(x)$. Within this energy-based, data-free paradigm, achieving comprehensive mode coverage emerges as a critical desideratum for faithfully representing the target distribution.

However, developing neural samplers that guarantee robust mode coverage under data-free constraints remains challenging. Prevailing approaches optimize objectives based on reverse metrics, such as

reverse KL divergence (Rezende & Mohamed, 2015; Luo et al., 2023) or variational bounds (Li & Turner, 2017; Shi et al., 2017; Zhang et al., 2019), which exhibit inherent mode-seeking behavior (Midgley et al., 2023b; He et al., 2024) and may miss smaller modes. Recent methods targeting coverage more directly, including FAB (Midgley et al., 2023b) and DiKL (He et al., 2024), either require computationally expensive MCMC within the training loop or rely on divergences related to reverse KL without strong theoretical guarantees for comprehensive mode discovery. Thus, existing neural samplers face a fundamental trade-off between computational efficiency and theoretical grounding when pursuing mode coverage.

In this work, we propose a principled approach rooted in diffusion process theory. We posit that objectives minimizing the forward KL divergence $\text{KL}(\mathbb{P}_r \,\|\, \mathbb{P}_\theta)$ between the true ($\mathbb{P}_r$) and learned diffusion ($\mathbb{P}_\theta$) path measures are conceptually better suited for mode coverage. This perspective arises because forward KL heavily penalizes assigning zero probability where the true process has mass, inherently driving the model to cover all modes (Song et al., 2021; Chen et al., 2022; 2023; Benton et al., 2023; Li et al., 2024). Minimizing this path KL is equivalent to a score matching objective (Song et al., 2021), which ideally requires expectations over the true marginal distributions $p_t(x)$. The central challenge, however, is that accessing samples from these true marginals $p_t(x)$ is precisely what is prohibited in the data-free setting.

To overcome this fundamental obstacle, we introduce *Importance Weighted Score Matching*, which operationalizes the optimization of the mode-covering score matching objective through importance sampling. While data-free methods naturally sample from an accessible proposal distribution $p_t^{\mathcal{B}}(x)$ derived from the neural sampler, our key contribution is correcting for the distribution mismatch through importance weighting. We derive a tractable Monte Carlo estimator for the importance weights $p_t(x)/p_t^{\mathcal{B}}(x)$ within a self-normalized importance sampling framework, bypassing intractable normalizing constants and relying solely on the energy function $E(x)$. This principled correction ensures unbiased gradient estimates despite sampling from $p_t^{\mathcal{B}}(x)$ rather than the true marginal $p_t(x)$. The neural sampler-induced proposal distribution adaptively improves during training, progressively approximating $p_t(x)$—a strategy well-established in adaptive importance sampling (Oh & Berger, 1992; Cappé et al., 2008; CORNUET et al., 2012) and recently applied to neural samplers (Gu et al., 2015; Akhound-Sadegh et al., 2024). We provide theoretical analysis of the bias and variance of our Monte Carlo estimators and the resulting practical loss.

We validate our method on diverse benchmarks: 2D Gaussian Mixture Models with increasing complexity (GMM-40, GMM-80, GMM-120) to assess mode coverage, and n-particle systems with symmetries: the 4-particle Double-Well potential, 13-particle Lennard-Jones cluster, and challenging 55-particle Lennard-Jones cluster (Köhler et al., 2020; Klein et al., 2023), to evaluate performance on physically relevant systems. Across multiple metrics including Wasserstein distances and Total Variation Distance, our approach consistently outperforms existing neural samplers , establishing new benchmarks.

## 2 BACKGROUND

**Problem Setup and Divergence Motivation.** We consider sampling from a target distribution $\pi(x) = \frac{\exp(-E(x))}{Z}$, $x \in \mathbb{R}^d$, defined by a known differentiable energy function $E(x)$ and an intractable normalization constant $Z$. Our goal is to train a neural sampler $p_\theta(x)$ to capture all significant modes in the *"data-free"* setting where samples from $\pi(x)$ are unavailable. We denote corresponding probability measures by $\Pi$ and $\mathbb{P}_\theta$. The choice of divergence for training is critical. Minimizing the *forward* KL divergence, $\text{KL}(\Pi \,\|\, \mathbb{P}_\theta) = \mathbb{E}_{p_t}[\log(\frac{d\Pi}{d\mathbb{P}_\theta})]$, encourages *mode-covering* behavior (Midgley et al., 2023b). In contrast, minimizing the *reverse* KL divergence, $\text{KL}(\mathbb{P}_\theta \,\|\, \Pi)$, often leads to *mode-seeking*, where the model may collapse onto a subset of modes (Bishop & Nasrabadi, 2006) (details in Appendix H).

**Importance Sampling.** Importance Sampling (IS) is a fundamental Monte Carlo technique to estimate $\mathbb{E}_{p(x)}[f(x)]$ when sampling from $p(x)$ is hard, but sampling from a proposal $q(x)$ is feasible (Tokdar & Kass, 2010). The expectation can be estimated by averaging $w(x_i)f(x_i)$ for samples $x_i \sim q(x)$, where $w(x) = p(x)/q(x)$ are importance weights: $\mathbb{E}_{p(x)}[f(x)] = \mathbb{E}_{q(x)}[w(x)f(x)] \approx \frac{1}{N}\sum_{i=1}^{N} w(x_i)f(x_i)$. When $p(x)$ and $q(x)$ are known only up to unnormalized densities $\tilde{p}(x), \tilde{q}(x)$, the weight $w(x) = (Z_q/Z_p)(\tilde{p}(x)/\tilde{q}(x))$ contains unknown constants. The

Self-Normalized Importance Sampling (SNIS) estimator $\hat{I}_{\text{SNIS}} = \frac{\sum_{i=1}^{N} \tilde{w}(x_i)f(x_i)}{\sum_{j=1}^{N} \tilde{w}(x_j)}$ uses unnormalized weights $\tilde{w}(x) = \tilde{p}(x)/\tilde{q}(x)$ to address this, but is known to be biased for finite sample sizes (Cardoso et al., 2022; Agapiou et al., 2017).

**Score-Based Diffusion Models.** Diffusion models (Ho et al., 2020; Song et al., 2021) learn to reverse a forward noise process defined by the stochastic differential equation (SDE):

$$dx_t = f(x_t, t)dt + g(t)dw_t, \tag{1}$$

which transforms samples $x_0 \sim \pi(x)$ to a simple prior $p_1$ at $t = 1$. The reverse-time SDE (Anderson, 1982):

$$dx_t = \left[ f(x_t, t) - g(t)^2 \nabla \log p_t(x_t) \right] dt + g(t)d\bar{w}_t, \tag{2}$$

generates samples from $\pi$ but requires the unknown score function $\nabla \log p_t(x_t)$. Score-based modeling trains a neural network $s_\theta(x_t, t)$ to approximate this score, enabling sampling via:

$$d\hat{x}_t = \left[ f(\hat{x}_t, t) - g(t)^2 s_\theta(\hat{x}_t, t) \right] dt + g(t)d\bar{w}_t. \tag{3}$$

Simulating Eq. 3 from $\hat{x}_1 \sim p_1$ backwards to $t = 0$ yields samples from the target distribution $\pi(x)$. Specific SDE formulations (VP-SDE, VE-SDE) are detailed in Appendix C.

**Sample Notations.** Lowercase letters denote random variables, with time indicated by subscripts (e.g., $x_t$ follows probability density $p_t$). Superscripts indicate samples, as in $x_t^{(i)}$ drawn from $p_t$. When unambiguous, $x_t$ also represents general samples in expressions like "$x_t \sim p_t$". We use $\{x_t^{(i)}\}_{i=1}^{N}$ for a set of $N$ samples, $\{x_t^{(i)}\}$ when sample size is unspecified, and $\{x_t\}_{t \in \mathcal{T}}$ for random variables indexed by time $t \in \mathcal{T}$.

**Related Works.** We leave the comprehensive review of related works in Appendix A due to the space limit.

## 3 METHODS

In this section, we detail our proposed Importance Weighted Score Matching method for training diffusion samplers in the data-free setting. We first derive the ideal score matching objective grounded in path KL divergence (section 3.1), introduce importance sampling to address the inaccessible true distribution (section 3.2), and develop tractable Monte Carlo estimators for the necessary components (section 3.3). Finally, we present the practical training algorithm utilizing self-normalized importance sampling (section 3.4).

### 3.1 THE PRINCIPLED SCORE MATCHING OBJECTIVE FROM PATH KL DIVERGENCE

Let $\mathbb{P}_r$ and $\mathbb{P}_\theta$ denote the probability measures over the space of continuous paths $\{x_t\}_{t \in [0,1]}$ induced by the true (Eq. 2) and model reverse SDEs (Eq. 3), respectively. As stated in Section 2, minimizing the forward KL divergence $\text{KL}(\mathbb{P}_r \,||\, \mathbb{P}_\theta)$ is conceptually desirable for ensuring comprehensive mode coverage. The equivalence between this path KL divergence and an equivalent score matching objective is established in the following proposition (Proofs provided in Appendix E.1).

**Proposition 1.** *Assume the diffusion coefficient $g(t) > 0$ for $t \in [0, 1]$ and standard regularity conditions hold. The KL divergence between the true reverse path measure $\mathbb{P}_r$ and the model reverse path measure $\mathbb{P}_\theta$ is given by:*

$$KL(\mathbb{P}_r \,||\, \mathbb{P}_\theta) = \mathbb{E}_{t \sim U(0,1)}\mathbb{E}_{x_t \sim p_t} \left[ \frac{g(t)^2}{2} \| s_\theta(x_t, t) - \nabla \log p_t(x_t) \|^2 \right] + C, \tag{4}$$

*where $C$ is a constant independent of $\theta$, and $U(0, 1)$ denotes the uniform distribution over $[0, 1]$.*

For simplicity in practical implementations, the weighting function $\frac{g(t)^2}{2}$ is often omitted (Ho et al., 2020), leading to the following ideal objective function derived from the path KL minimization perspective:

$$L_{\text{ideal}}(\theta) = \mathbb{E}_{t \sim U(0,1)}\mathbb{E}_{x_t \sim p_t} \left[ \| s_\theta(x_t, t) - \nabla \log p_t(x_t) \|^2 \right] := \mathbb{E}_{t \sim U(0,1)} \left[ L_{\text{ideal}}(\theta, t) \right]. \tag{5}$$

While the score matching objectives in Eq. 5 are standard in generative modeling when target samples are available (Ho et al., 2020; Song et al., 2021), optimizing it directly in the data-free setting,

however, faces two major obstacles: **(1)** Inaccessibility of $p_t$ samples: The expectation $\mathbb{E}_{x_t \sim p_t}[\cdot]$ cannot be computed via standard Monte Carlo, as in the data-free setting we lack samples $x_0 \sim \pi(x)$ needed to generate samples $x_t \sim p_t$ via the forward SDE (Eq. 1); **(2)** Unknown score function $\nabla \log p_t(x_t)$: The target of the inner squared error, the true score $\nabla \log p_t(x_t)$, is itself unknown because $p_t$ depends on $\pi$ with unknown normalization constant $Z$. Standard techniques like denoising score matching (Vincent, 2011), which bypass explicit knowledge of $\nabla \log p_t$ when $p_0$ samples are available, cannot be applied here. To operationalize the optimization of Eq. 5 under these constraints, we next introduce importance sampling to address the inaccessible expectation $\mathbb{E}_{x_t \sim p_t}[\cdot]$.

## 3.2 BRIDGING DISTRIBUTIONS: THE SAMPLER-INDUCED PROPOSAL DISTRIBUTION

To facilitate optimization for $L_{\text{ideal}}$ in Eq. 5, we utilize importance sampling (IS) to construct a weighted estimator based on samples from an accessible proposal distribution. A crucial element for effective importance sampling is the choice of a proposal distribution which should have significant overlap with the target marginal $p_t(x_t)$. In the iterative training context, where the score model $s_\theta$ is continually updated, a natural and adaptive proposal arises from the sampler's own generated history. We maintain a replay buffer $\mathcal{B} = \{x_0^{(i)}\}$ containing past samples generated by simulating the model reverse SDE (Eq. 3) using the score network $s_\theta$ across the training. Let $p_0^{\mathcal{B}}(x_0)$ denote the empirical distribution represented by the samples currently stored in buffer $\mathcal{B}$, which serves as the foundation for our proposal distribution. To ensure that the support of $p_0^{\mathcal{B}}$ adequately covers the target distribution, we augment $\mathcal{B}$ with a portion of coverage prior samples, such as those drawn from Gaussian or uniform distributions.

Applying the known forward diffusion kernel $p_{t|0}(x_t|x_0)$ to samples drawn from the buffer induces the time-$t$ marginal proposal distribution: $p_t^{\mathcal{B}}(x_t) = \int p_{t|0}(x_t|x_0)p_0^{\mathcal{B}}(x_0)dx_0$. Sampling $x_t \sim p_t^{\mathcal{B}}$ is straightforward: first sample $x_0 \sim p_0^{\mathcal{B}}$ and then sample $x_{t|0} \sim p_{t|0}(\cdot|x_0)$. As the sampler $s_\theta$ improves over training iterations, the buffer $\mathcal{B}$ is refreshed with progressively higher-quality samples, causing $p_0^{\mathcal{B}}$ (and consequently $p_t^{\mathcal{B}}$) to progressively better approximate the true distributions $\pi$ and $p_t$. Then applying the IS principle to the ideal objective $L_{\text{ideal}}$ (Eq. 5) with $p_t^{\mathcal{B}}$ as the proposal distribution yields:

$$L_{\text{ideal}}(\theta) = \mathbb{E}_{t \sim U(0,1)} \mathbb{E}_{x_t \sim p_t^{\mathcal{B}}} \left[ w(x_t) \| s_\theta(x_t, t) - \nabla \log p_t(x_t) \|^2 \right], \text{ where } w(x_t) = \frac{p_t(x_t)}{p_t^{\mathcal{B}}(x_t)}. \quad (6)$$

It is worth noting that iDEM (Akhound-Sadegh et al., 2024) adopts a similar loss but with $w(x_t) = 1$ for all $x_t$, optimizing the score matching objective under the proposal distribution $p_t^{\mathcal{B}}$ rather than the true $p_t$. While this serves as a practical heuristic objective that avoids computing importance weights, it lacks theoretical guarantees for convergence to the correct score function, particularly when $p_t^{\mathcal{B}}$ significantly deviates from $p_t$. We provide a detailed discussion of the relationships between our method, iDEM, and reverse KL objective in Appendix F. While this reformulation shifts the expectation to the accessible proposal $p_t^{\mathcal{B}}$, we still face the challenges of estimating the intractable score $\nabla \log p_t(x_t)$ and the importance weight ratio $w(x_t)$.

## 3.3 REALIZING THE BRIDGE: TRACTABLE ESTIMATORS FOR SCORES AND WEIGHTS

In this section, we address the practical estimation of the two key unknown quantities: the true score function $\nabla \log p_t(x_t)$ and the importance weight ratio $w(x_t)$.

**Target Score Estimation.** As direct computation is intractable, we employ a previously established Monte Carlo estimator for score function (Akhound-Sadegh et al., 2024) and gradient of potential function (Huang et al., 2024), suitable for energy-based settings. Specifically, for diffusion processes involving Gaussian kernels, such as the VE-SDE where $p_{t|0}(x_t|x_0) = \mathcal{N}(x_t; x_0, \sigma_t^2 I)$, the score can be approximated by:

$$\nabla \log p_t(x_t) \approx S_L(x_t, t) := \nabla_{x_t} \log \sum_{i=1}^{L} \exp(-E(x_{0|t}^{(i)})), \text{ where } \{x_{0|t}^{(i)}\}_{i=1}^{L} \sim \mathcal{N}(x_0; x_t, \sigma_t^2 I).$$
$$(7)$$

In fact, the estimator $S_L(x_t, t)$ can be viewed as a target score identity (TSI) (Bortoli et al., 2024) estimate using Gaussian distributions as proposals, which provides lower-variance and more accurate

estimates particularly when the noise level is small. The detailed derivation of this score estimator, its estimation error analysis, and the transformation of general SDEs into the VE-SDE form via suitable change of variables are provided in Appendix D. Given this established generality, our subsequent discussions and theoretical developments will primarily adopt the VE-SDE framework, unless explicitly stated otherwise. Additionally, with this estimator, we can perform direct sampling using the reverse SDE in Equation 2. Discussions regarding the effectiveness and efficiency of this sampling approach are presented in Specifically, Appendix I.

While prior work Akhound-Sadegh et al. (2024) has examined the bias of $S_L(x_t, t)$ (Eq. 7) under restrictive sub-Gaussian assumptions on the energy function $E(x)$, we aim to provide analysis on the bias and variance analysis under potentially broader conditions here.

**Proposition 2.** *Consider a diffusion process governed by a VE-SDE. Assume $E(x)$ is bounded below and $\nabla E(x)$ is Lipschitz continuous with bounded norm. For a given state $x_t$ and $t > 0$, the bias and variance of the score estimator $S_L(x_t, t)$ in Eq. 7 satisfy:*

$$\|\mathbb{E}[S_L(x_t, t)] - \nabla \log p_t(x_t)\| \leq \frac{c(x_t, t)}{\sqrt{L}}, \ Var[S_L(x_t, t)] \leq \frac{c^2(x_t, t)}{L}, \tag{8}$$

*where the expectation and variance are taken over $\{x_{0|t}^{(i)}\}_{i=1}^L \sim \mathcal{N}(x_0; x_t, \sigma_t^2 I)$ and $c(x_t, t)$ is a constant that depends on $x_t$ and $t$. Proofs are provided in Appendix E.2.*

**Importance Weight Estimation.** The challenge lies in estimating the ratio of intractable marginal densities $w(x_t) = p_t(x_t)/p_t^{\mathcal{B}}(x_t)$. We construct estimators for quantities proportional to the numerator and the denominator separately.

*Numerator Estimation $p_t(x_t)$:* The true marginal density is given by $p_t(x_t) = \int p_{t|0}(x_t|x_0)\pi(x_0)dx_0 = Z^{-1} \int p_{t|0}(x_t|x_0)e^{-E(x_0)}dx_0 := Z^{-1}I(x_t, t)$. We can estimate $I(x_t, t) = \int p_{t|0}(x_t|x_0)e^{-E(x_0)}dx_0$ using Monte Carlo method. For VE-SDE, where the forward transition kernel is given by $p_{t|0}(x_t|x_0) = \mathcal{N}(x_t; x_0, \sigma_t^2 I)$, we leverage the symmetry property of Gaussian density: $\mathcal{N}(x_t; x_0, \sigma_t^2 I) = \mathcal{N}(x_0; x_t, \sigma_t^2 I)$, which enables us to compute the integral $I(x_t, t)$ as follows:

$$I(x_t, t) = Z \cdot p_t(x_t) = \int \mathcal{N}(x_t; x_0, \sigma_t^2 I)e^{-E(x_0)}dx_0 = \int \mathcal{N}(x_0; x_t, \sigma_t^2 I)e^{-E(x_0)}dx_0$$

$$= \mathbb{E}_{x_0 \sim \mathcal{N}(x_0; x_t, \sigma_t^2 I)}\left[e^{-E(x_0)}\right] \approx \frac{1}{K}\sum_{i=1}^K \exp(-E(x_{0|t}^{(i)})) := N_K(x_t), \tag{9}$$

where $\{x_{0|t}^{(i)}\}_{i=1}^K$ denotes a set of $K$ i.i.d. samples drawn from the distribution $\mathcal{N}(x_0; x_t, \sigma_t^2 I)$. In fact, this estimator can be viewed as importance sampling with a Gaussian proposal distribution, exhibiting similar error characteristics to the estimator introduced in Eq. 7.

*Denominator Estimation $p_t^{\mathcal{B}}(x_t)$:* The proposal density $p_t^{\mathcal{B}}(x_t) = \int p_{t|0}(x_t|x_0)p_0^{\mathcal{B}}(x_0)dx_0$ can be directly estimated via Monte Carlo method using samples from the buffer $\mathcal{B}$. By drawing $M$ samples $\{x_0^{(j)}\}_{j=1}^M$ from the empirical distribution $p_0^{\mathcal{B}}$, we formulate the estimator as:

$$p_t^{\mathcal{B}}(x_t) \approx D_M(x_t) = \frac{1}{M}\sum_{i=1}^M p_{t|0}(x_t|x_0^{(i)}), \tag{10}$$

where each term evaluates the forward transition kernel at $x_t$ conditioned on sampled buffer samples.

*Self-Normalization Estimator:* Given the estimators $N_K(x_t)$ (Eq. 9) and $D_M(x_t)$ (Eq. 10), the ratio of our estimators provides an estimate proportional to this true weight $w(x_t)$ up to the unknown normalization constant $Z$:

$$\tilde{w}(x_t) = \frac{N_K(x_t)}{D_M(x_t)} = \frac{\frac{1}{K}\sum_{i=1}^K \exp(-E(x_{0|t}^{(i)}))}{\frac{1}{M}\sum_{i=1}^M p_{t|0}(x_t|x_0^{(i)})} \approx Z \cdot w(x_t). \tag{11}$$

While theoretically we could substitute $\tilde{w}(x_t)/Z$ for $w(x_t)$ in Eq. 6 and omit $Z$ during training since the constant factor does not affect optimization objectives, in practice $\tilde{w}(x_t)$ involves sums

of exponential terms for energy-based distributions with large dynamic ranges, which can lead to severe numerical instability issues. To eliminate the intractable dependence, we employ the self-normalized importance sampling (SNIS) technique to obtain the SNIS weights computed over a batch $\mathcal{S} = \{x_t^{(s)}\}_{s=1}^S$ sampled from the proposal distribution $p_t^{\mathcal{B}}$:

$$\tilde{w}_{\text{SNIS}}(x_t^{(s)}) = \frac{\tilde{w}(x_t^{(s)})}{\sum_j \tilde{w}(x_t^{(j)})}, \tag{12}$$

which inherently cancels unknown normalization factors common to all weights within an expectation estimate, ensuring numerical stability while preserving the relative importance of each sample. Combining the developed estimator $S_L(x_t, t)$ in Eq. 7, we obtain the SNIS based loss function:

$$L_{\text{SNIS}}(\theta, t) = \sum_{s=1}^S \tilde{w}_{\text{SNIS}}(x_t^{(s)}) \| s_\theta(x_t^{(s)}, t) - S_L(x_t^{(s)}) \|^2, \text{ where } \{x_t^{(s)}\}_{s=1}^S \sim p_t^{\mathcal{B}}. \tag{13}$$

Given the multiple Monte Carlo approximations in this empirical objective, we analyze the bias and variance characteristics of this estimator in the following proposition.

**Proposition 3.** *Let $L^*(\theta, t) := \mathbb{E}_{x_t \sim p_t}[\|s_\theta(x_t, t) - \nabla_{x_t} \log p_t(x_t)\|^2]$ be the target loss. Consider the score estimator $S_L(x_t, t)$ (Eq. 7), a data batch $\mathcal{S} = \{x_t^{(s)}\}_{s=1}^S$ from $p_t^{\mathcal{B}}(x_t)$, the unnormalized importance weight estimate $\tilde{w}(x_t)$ (Eq. 11), and the SNIS loss estimator $L_{SNIS}(\theta, t)$ (Eq. 13). Under mild regularity conditions, for a fixed $t > 0$, the bias and mean squared error of $L_{SNIS}(\theta, t)$ satisfy:*

$$|\mathbb{E}[L_{SNIS}(\theta, t)] - L^*(\theta, t)| \leq \frac{K_1 V_{\tilde{w}}}{S} + f(\frac{1}{\sqrt{L}}),$$

$$\mathbb{E}[(L_{SNIS}(\theta, t) - L^*(\theta, t))^2] \leq \frac{K_1^2 V_{\tilde{w}}}{36S} + f(\frac{1}{\sqrt{L}}) \left( \frac{2K_1 V_{\tilde{w}}}{S} + f(\frac{1}{\sqrt{L}}) \right),$$

*where $f(x) = K_2 x + K_3 x^2$, $V_{\tilde{w}}$ reflects the variability of the importance weight estimates. The constants $K_1, K_2, K_3$ are independent of the sample sizes $S$ and $L$. Detailed conditions and expressions for these constants are provided in the proof (Appendix E.3).*

### 3.4 THE IMPORTANCE WEIGHTED SCORE MATCHING ALGORITHM

As derived in Section 3.2, employing importance sampling to address the inaccessible expectation in the ideal score matching objective (Eq. 5) yields the following formulation based on the proposal distribution $p_t^{\mathcal{B}}$ and true importance weight $w(x_t)$:

$$L_{\text{ideal}}(\theta, t) = \mathbb{E}_{x_0 \sim p_0^{\mathcal{B}}, \, x_t \sim p_{t|0}(\cdot|x_0)} \left[ w(x_t) \| s_\theta(x_t, t) - \nabla \log p_t(x_t) \|^2 \right]. \tag{14}$$

We approximate the intractable components using the SNIS weights $\tilde{w}_{\text{SNIS}}$ (Eq. 12) for the importance weight $w(x_t)$ and the Monte Carlo estimator $S_L(x_t, t)$ (Eq. 7) for the true score function $\nabla \log p_t(x_t)$. This leads to our practical objective function as follows: For a given sample $x_0 \sim p_0^{\mathcal{B}}$ and time parameter $t \sim U(0, 1)$, we first generate a batch of samples $\{x_{t|0}^{(s)}\}_{s=1}^S \sim p_{t|0}(\cdot|x_0)$. The empirical estimate of the $x_0$-conditioned objective, denoted $L_{\text{SNIS}}(\theta, t|x_0)$, is then computed as:

$$L_{\text{SNIS}}(\theta, t|x_0) = \sum_{s=1}^S \tilde{w}_{\text{SNIS}}(x_{t|0}^{(s)}) \| s_\theta(x_{t|0}^{(s)}, t) - S_L(x_{t|0}^{(s)}, t) \|^2. \tag{15}$$

Then we obtain the empirical estimate of $L_{\text{ideal}}(\theta, t)$ for a batch $\{x_0^{(b)}\}_{b=1}^B \sim p_0^{\mathcal{B}}$, denoted $L_{\text{batch}}(\theta, t)$, is computed as:

$$L_{\text{batch}}(\theta, t) = \frac{1}{B} \sum_{b=1}^B L_{\text{SNIS}}(\theta, t|x_0^{(b)}) \approx L_{\text{ideal}}(\theta, t), \tag{16}$$

which serves as the trainable objective function minimized via stochastic gradient descent. See Appendix B for the algorithm (Algorithm 1) and practical considerations during implementation.

Table 1: Comparison of sampling methods across Gaussian mixture models of varying complexity. Metrics include 1-Wasserstein ($\mathcal{W}_1$) and 2-Wasserstein ($\mathcal{W}_2$) distances, energy total variation distance (E-TVD), and sample level total variation distance (S-TVD). Values reported as mean$\pm$std across 3 random seeds.

| Method | GMM-40 | | | | GMM-80 | | | | GMM-120 | | | |
|---|---|---|---|---|---|---|---|---|---|---|---|---|
| | $\mathcal{W}_1$ | $\mathcal{W}_2$ | E-TVD | S-TVD | $\mathcal{W}_1$ | $\mathcal{W}_2$ | E-TVD | S-TVD | $\mathcal{W}_1$ | $\mathcal{W}_2$ | E-TVD | S-TVD |
| PIS | $5.98_{\pm0.1}$ | $7.18_{\pm0.2}$ | $0.56_{\pm0.0}$ | $0.68_{\pm0.0}$ | $36.14_{\pm0.2}$ | $41.85_{\pm0.3}$ | $0.64_{\pm0.0}$ | $0.86_{\pm0.0}$ | $55.72_{\pm0.6}$ | $63.75_{\pm0.6}$ | $0.61_{\pm0.0}$ | $0.89_{\pm0.0}$ |
| FAB | $4.52_{\pm5.0}$ | $5.71_{\pm5.3}$ | $0.35_{\pm0.2}$ | $0.53_{\pm0.3}$ | $13.02_{\pm5.1}$ | $17.26_{\pm4.9}$ | $0.49_{\pm0.1}$ | $0.70_{\pm0.2}$ | $21.67_{\pm8.6}$ | $27.56_{\pm11.6}$ | $0.58_{\pm0.1}$ | $0.79_{\pm0.1}$ |
| DiKL | $2.68_{\pm0.2}$ | $4.84_{\pm0.3}$ | $0.09_{\pm0.0}$ | $0.35_{\pm0.0}$ | $10.16_{\pm0.1}$ | $15.26_{\pm0.1}$ | $0.23_{\pm0.0}$ | $0.54_{\pm0.0}$ | $27.91_{\pm0.7}$ | $39.91_{\pm0.5}$ | $0.26_{\pm0.0}$ | $0.62_{\pm0.0}$ |
| pDEM | $3.44_{\pm1.1}$ | $5.62_{\pm1.4}$ | $0.10_{\pm0.0}$ | $0.30_{\pm0.0}$ | $6.50_{\pm0.9}$ | $9.88_{\pm0.5}$ | $0.40_{\pm0.1}$ | $0.54_{\pm0.2}$ | $15.27_{\pm2.5}$ | $20.36_{\pm2.3}$ | $0.53_{\pm0.0}$ | $0.69_{\pm0.1}$ |
| iDEM | $2.52_{\pm1.3}$ | $4.40_{\pm2.1}$ | $0.07_{\pm0.0}$ | $0.23_{\pm0.0}$ | $6.04_{\pm2.7}$ | $10.18_{\pm3.3}$ | $0.13_{\pm0.0}$ | $0.26_{\pm0.1}$ | $9.23_{\pm0.8}$ | $14.94_{\pm1.0}$ | $0.32_{\pm0.0}$ | $0.41_{\pm0.0}$ |
| Ours | $\mathbf{1.43}_{\pm0.6}$ | $\mathbf{3.12}_{\pm1.3}$ | $\mathbf{0.05}_{\pm0.0}$ | $\mathbf{0.19}_{\pm0.0}$ | $\mathbf{3.21}_{\pm0.4}$ | $\mathbf{6.58}_{\pm0.7}$ | $\mathbf{0.12}_{\pm0.0}$ | $\mathbf{0.21}_{\pm0.0}$ | $\mathbf{5.05}_{\pm0.9}$ | $\mathbf{9.90}_{\pm1.2}$ | $\mathbf{0.23}_{\pm0.0}$ | $\mathbf{0.30}_{\pm0.0}$ |

Figure 1: Visualization of samples from different methods across GMM benchmarks.

## 4 EXPERIMENTS

In this section, we evaluate our proposed method to assess its effectiveness in enhancing mode coverage and sample quality in the data-free setting compared to existing approaches.

**Benchmarks.** We evaluate sampler performance on a diverse suite of target distributions. For mode-covering assessment, particularly under increasing complexity, we use 2D Gaussian Mixture Models (GMMs) with 40, 80, and 120 modes (GMM-40, GMM-80, GMM-120), presenting increasing challenges beyond standard GMM-40 used in previous works (Akhound-Sadegh et al., 2024; He et al., 2024). For scientifically relevant tasks with complex, multi-modal energy landscapes and inherent SE(3) and permutation symmetries, we use $n$-particle systems: the 4-particle Double-Well (DW-4), the 13-particle Lennard-Jones (LJ-13), and the highly challenging 55-particle LJ-55 cluster (Köhler et al., 2020; 2023). Reference samples are from ground truth (GMMs) or long-run MCMC (particle systems). Further details on each benchmark distribution are provided in Appendix G.1.

**Metrics.** We report standard metrics to evaluate sampler performance. Overall distributional similarity and mode sensitivity are measured by the 1-Wasserstein ($\mathcal{W}_1$) and 2-Wasserstein ($\mathcal{W}_2$) distances (Villani & Villani, 2009; Panaretos & Zemel, 2019). We also assess the energy landscape coverage using Total Variation Distance on log-energy histograms (E-TVD) for all benchmarks (Levin & Peres, 2017). Benchmark-specific TVD metrics capture spatial or geometric accuracy: 2D sample space TVD (S-TVD) for GMMs, and pairwise interatomic distance TVD (D-TVD) for particle systems. Further details of these metrics are provided in Appendix G.2.

**Baselines.** We evaluate the efficacy of our proposed method relative to several established baseline algorithms. The comparative methods encompass FAB (Midgley et al., 2023b), PIS (Zhang & Chen, 2022), iDEM (Akhound-Sadegh et al., 2024), and DiKL (He et al., 2024). Although we attempted to include DDS (Vargas et al., 2023a), DIS (Berner et al., 2022), and CMCD (Vargas et al., 2023b) using publicly available implementations , these methods yielded poor performance on our benchmarks. We also exclude LRDS (Noble et al., 2024) as it requires privileged access to target distribution statistics. Comprehensive descriptions of these baseline methodologies and their operational principles are

Table 2: Comparison of sampling methods across particle system benchmarks. Metrics include 1-Wasserstein ($\mathcal{W}_1$) and 2-Wasserstein ($\mathcal{W}_2$) distances, energy total variation distance (E-TVD), and distance total variation distance (D-TVD). Values reported as mean±std across 3 random seeds. The symbol $*$ indicates that calculation results diverged.

| Method | DW-4 | | | | LJ-13 | | | | LJ-55 | | | |
|---|---|---|---|---|---|---|---|---|---|---|---|---|
| | $\mathcal{W}_1$ | $\mathcal{W}_2$ | E-TVD | D-TVD | $\mathcal{W}_1$ | $\mathcal{W}_2$ | E-TVD | D-TVD | $\mathcal{W}_1$ | $\mathcal{W}_2$ | E-TVD | D-TVD |
| FAB | 1.33±0.0 | 1.58±0.0 | 0.09±0.1 | 0.04±0.0 | 4.74±0.0 | 4.77±0.0 | 0.91±0.0 | 0.25±0.0 | 18.48±1.6 | 18.48±1.6 | $*$ | 0.28±0.1 |
| DiKL | 1.38±0.1 | 1.63±0.1 | 0.10±0.1 | 0.09±0.1 | 4.02±0.0 | 4.03±0.0 | 0.17±0.0 | **0.04**±0.0 | $*$ | $*$ | $*$ | $*$ |
| iDEM | 1.32±0.0 | 1.58±0.0 | 0.08±0.0 | 0.05±0.0 | 4.01±0.1 | 4.03±0.1 | 0.22±0.1 | 0.06±0.0 | 16.22±0.0 | 16.22±0.0 | 0.99±0.0 | 0.11±0.0 |
| Ours | **1.31**±0.0 | **1.57**±0.0 | **0.05**±0.0 | 0.04±0.0 | **3.86**±0.0 | **3.87**±0.0 | **0.15**±0.0 | 0.08±0.0 | **15.67**±0.0 | **15.68**±0.0 | **0.87**±0.0 | **0.05**±0.0 |

deferred to Appendix G.3. Additional implementation details, including network configurations, optimization procedures, and evaluation protocols, are provided in Appendix G.4.

## 4.1 RESULTS ON GMMs

Table 1 and Figure 1 present the performance of our method against baseline approaches on GMM-40, GMM-80, and GMM-120. As shown in Figure 1, baseline methods demonstrate various limitations: PIS fails to adequately separate the modes and produces diffuse, overlapping samples; FAB generates distinct but distorted mode structures with spurious connections between clusters; DiKL improves mode separation but still misses some modes especially in the higher-complexity GMM-80 and GMM-120 settings; iDEM achieves better mode coverage but with less precise mode shapes and noticeably noisier sample distributions compared to our approach. In contrast, our method consistently captures the complex multi-modal structure across all three GMM benchmarks, with sample distributions closely resembling the ground truth in both mode coverage and precision, accurately representing the underlying density without introducing noise or distortion. The quantitative results in Table 1 confirm these observations, showing the excellent performance of our method. On GMM-40, our approach demonstrates a clear advantage, reducing the Wasserstein distances by 43% ($\mathcal{W}_1$) and 29% ($\mathcal{W}_2$) compared to the next best method (iDEM). The performance advantage is more pronounced on the more challenging GMM-80 and GMM-120 benchmarks. On GMM-80, our method reduces $\mathcal{W}_1$ and $\mathcal{W}_2$ by 47% and 35% respectively compared to iDEM, while on GMM-120, reductions are 45% and 34%. Consistently low S-TVD values further demonstrate our method's ability to accurately capture the underlying sample space probability density. These results highlight the scalability of our approach to increasingly complex multi-modal distributions where baseline methods struggle.

## 4.2 RESULTS ON PARTICLE SYSTEMS

Table 2 presents our method's performance on particle system benchmarks: DW-4, LJ-13, and the challenging LJ-55. The Wasserstein distances ($\mathcal{W}_1$ and $\mathcal{W}_2$) provide a comprehensive measure of the overall sample distribution quality. For the DW-4 system, our method achieves modest improvements compared to the next best performer iDEM . For the more complex LJ-13 system, the improvement becomes more pronounced with our method achieving $\mathcal{W}_1 = 3.86$ and $\mathcal{W}_2 = 3.87$, outperforming iDEM ($\mathcal{W}_1 = 4.01$, $\mathcal{W}_2 = 4.03$) and substantially surpassing DiKL. In the highly challenging LJ-55 benchmark, our method again demonstrates superior performance ($\mathcal{W}_1 = 15.67$, $\mathcal{W}_2 = 15.68$) vs iDEM ($\mathcal{W}_1 = \mathcal{W}_2 = 16.22$), while DiKL's calculations diverge entirely and FAB shows significantly worse performance. E-TVD and energy distributions assess energy landscape capture. For DW-4, our method's E-TVD (0.05) is the lowest among tested methods (range 0.08-0.10). The LJ-13 system presents a more significant challenge, particularly for FAB, which produces a dramatically shifted distribution, resulting in a high E-TVD of 0.91, while DiKL (0.17) and iDEM (0.22) capture the energy distribution more accurately but with noticeable deviations in peak height and tail behavior. Our method achieves the closest match to the ground truth energy distribution shape with an E-TVD of 0.15. For LJ-55, DiKL diverges and FAB is unstable; our method (E-TVD 0.87) and iDEM (0.99) show shifts relative to the reference, but our distribution shape better resembles the reference. D-TVD captures structural accuracy. For DW-4, our method matchs FAB for the lowest D-TVD (0.04). On LJ-13, DiKL achieves the lowest D-TVD (0.04), while our method (0.08) better captures critical peak heights than FAB (0.25). For LJ-55, our method demonstrates superior structural accuracy (0.05 D-TVD) compared to iDEM (0.11) and FAB (0.28). Further visualizations for interatomic distance distribution are in Figure 5 in Appendix G.5

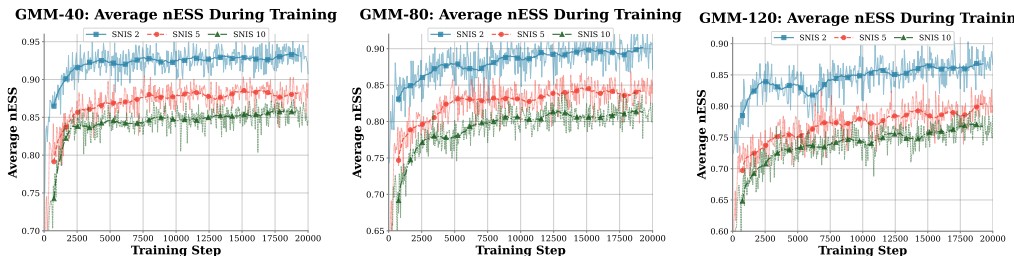

Figure 2: Normalized Effective Sample Size (nESS) comparison across different SNIS configurations for GMM benchmarks during training.

### 4.3 EFFECTS OF IMPORTANCE WEIGHT CORRECTION

**Effective Sample Size Analysis.** To understand the behavior of our importance weighting scheme, we monitor the normalized Effective Sample Size (nESS), defined as: $\text{nESS} = 1/\sum_{s=1}^{S} \tilde{w}_{\text{SNIS}}^2(x_t^{(s)}) \in [1/S, 1]$, where $\tilde{w}_{\text{SNIS}}$ defined in Eq. 12. The nESS quantifies the weight distribution balance in SNIS, with values close to 1 indicating uniform weights across samples and values near $1/S$ indicating that one sample dominates. Figure 2 presents the nESS evolution during training across GMM-40, GMM-80, and GMM-120 benchmarks for our method with $S \in \{2, 5, 10\}$ proposal samples. We observe that: (1) The nESS rapidly increases during early training and stabilizes, indicating that the proposal distribution $p_t^{\mathcal{B}}$ progressively adapts to the target; (2) As problem complexity increases from GMM-40 to GMM-120, the nESS values show a slight decrease, yet all configurations maintain relatively balanced weights with nESS $> 0.75$; (3) The stable nESS throughout training demonstrates consistent importance weight estimation without degeneracy. We provide additional analysis of nESS behavior in high-dimensional settings in Appendix G.6.

**Training and Evaluation Performance.** The baseline method sets $\tilde{w}_{\text{SNIS}}(x_t^{(s)}) = 1$. Figure 3 illustrates the substantial benefits of SNIS on GMM-40. In the left panel, all SNIS variants (2, 5, 10 proposal samples) achieve significantly lower training losses than the baseline (e.g., SNIS-5/10 around 2.2, SNIS-2 around 3.0, vs baseline 6.2). SNIS variants also demonstrate faster convergence and greater training stability, with SNIS-10 showing the smoothest trajectory, aligning with the results in Proposition 3. The right panel demonstrates that improved training translates to enhanced sample quality. SNIS-2 significantly reduces Wasserstein distances over baseline ($\mathcal{W}_1$: 3.9 vs 4.3, $\mathcal{W}_2$: 6.3 vs 7.6) with minimal cost (1.2 vs 1.0 hrs). SNIS-5 further

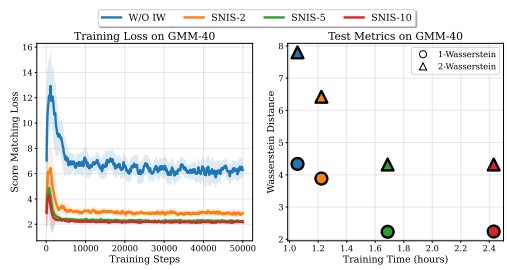

Figure 3: Comparison of training and testing performance on the GMM-40. Left: Training loss trajectories for different importance sampling strategies. Right: Evaluation of trained models using $\mathcal{W}_1$ and $\mathcal{W}_2$ metrics.

improves metrics ($\mathcal{W}_1 = 2.2, \mathcal{W}_2 = 4.3$) at 1.7 hrs. SNIS-10 yields nearly identical metrics to SNIS-5, indicating performance plateaus. These results demonstrate that a relatively small computational overhead from SNIS can yield substantial improvements in both training stability and sampling quality, with diminishing returns beyond a certain point. Similar performance trends were observed across other GMM benchmark distributions, with detailed results presented in Appendix G.7.

## 5 CONCLUSIONS

In this work, we introduced *Importance Weighted Score Matching*, a principled diffusion-based method for training neural samplers from unnormalized densities with comprehensive mode coverage. By approximating a mode-covering objective analogous to forward KL divergence through tractable importance sampling, our approach overcomes the inherent mode-seeking behavior of data-free training. Extensive experiments on multi-modal GMMs and particle systems demonstrate state-of-the-art performance across all metrics, validating our method's superior mode coverage and distributional accuracy. Future directions and limitations are discussed in Appendix J.

## REPRODUCIBILITY STATEMENT

To ensure the reproducibility of our results, we provide comprehensive implementation details throughout the paper and appendices. Complete descriptions of all benchmark distributions, including their energy functions and parameter settings, are provided in Appendix G.1. Detailed formulations of all evaluation metrics are given in Appendix G.2. For baseline comparisons, we utilized official implementations from the original authors where available, with specific repository links and version details documented in Appendix G.4. Our experimental configuration, including network architectures (MLPs for GMMs, EGNNs for particle systems), hyperparameters (learning rates, noise schedules, SNIS sample quantities, buffer sizes), and optimization procedures, is fully specified in Appendix G.4. We maintain consistency with established experimental protocols from prior work, particularly following the setup from iDEM for fair comparison. All experiments were conducted on single NVIDIA A40 GPUs, with three independent runs using different random seeds to ensure statistical reliability. The specific random seeds used for data generation and model initialization are documented to enable exact replication. Our implementation code, including training scripts and evaluation pipelines, will be made publicly available upon publication.

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

## A    RELATED WORKS

**Neural Samplers.**    Several neural sampling approaches (Marzouk et al., 2016; Noé et al., 2019) have emerged for approximating complex target distributions. Score-based methods (Li & Turner, 2017; Zhang et al., 2019; Song et al., 2020; Luo et al., 2023) offer theoretical guarantees but struggle with multi-modality, while recent advances like Denoising Diffusion Samplers (Vargas et al., 2023a) and Iterated Denoising Energy Matching (Akhound-Sadegh et al., 2024) improve performance but require expensive numerical integration. Flow-based approaches (Midgley et al., 2023b; Albergo et al., 2019; Gabrié et al., 2022), achieve better mode coverage but demand specialized invertible architectures—limitations that extend to their equivariant counterparts (Köhler et al., 2020; Midgley et al., 2023a; Klein et al., 2023). Diffusion process methods (Berner et al., 2022; Zhang & Chen, 2022; Albergo & Vanden-Eijnden, 2024; Vargas et al., 2023b; Richter & Berner, 2023; Noble et al., 2024) establish connections to stochastic optimal control but typically are based on Log-variance objectives which involve computationally expensive path simulations. Hybrid techniques combining normalizing flows with MCMC (Wu et al., 2020; Geffner & Domke, 2021; Thin et al., 2021), Sequential Monte Carlo methods (Chen et al., 2024), and replay buffer strategies (Midgley et al., 2023b; Akhound-Sadegh et al., 2024) have shown promise in balancing exploration and exploitation. Concurrent work (Havens et al., 2025) explores efficient training of neural samplers via adjoint methods, offering a complementary approach to computational efficiency.

**Importance Sampling.**    Importance Sampling (IS) stands as a fundamental Monte Carlo technique widely employed for estimating expectations. This is achieved by drawing samples from a more tractable proposal distribution and subsequently re-weighting these samples by the importance ratio (Bucklew, 2004; Robert & Casella, 2013). In cases of the target distribution is accessible through an unnormalized density, Self-Normalized Importance Sampling (SNIS) (Owen, 2013b; Robert & Casella, 2013) offers a practical solution. While the SNIS estimator is known to be asymptotically consistent, it inherently exhibits bias for finite sample sizes (Owen, 2013b). To mitigate the variance issue of IS, Adaptive Importance Sampling methods (Oh & Berger, 1992; Cappé et al., 2008; CORNUET et al., 2012) have been developed. These techniques iteratively refine the proposal distribution $q(x)$ throughout the sampling or optimization process. In recent years, the principles of Importance Sampling have been successfully applied to the training of neural samplers (Gu et al., 2015; Müller et al., 2019; Midgley et al., 2023b; Noé et al., 2019; Sanokowski et al., 2025). Early work introduced Neural Adaptive Sequential Monte Carlo to adapt SMC proposals using neural networks by minimizing the inclusive KL divergence (Gu et al., 2015). More recently, FAB leveraged Annealed Importance Sampling with an $\alpha$-divergence objective to train normalizing flows on unnormalized densities (Midgley et al., 2023b). Building on this, Scalable Discrete Diffusion Samplers (SDDS) (Sanokowski et al., 2025) explored training discrete diffusion models on unnormalized distributions, proposing a method based on Self-Normalized Neural Importance Sampling for scalable and unbiased forward KL gradient estimation.

**Off-policy Training.**    Off-policy training is a fundamental paradigm in reinforcement learning (RL) where an agent learns to evaluate or improve a policy different from the one currently used to generate data (Sutton & Barto, 2018). The ability to learn from data generated by arbitrary or older policies offers significant advantages, including increased data efficiency by reusing past experiences, the capacity to learn about optimal policies while exploring safely with a more stochastic behavior policy, and the potential to combine data from multiple sources or agents. Key off-policy RL algorithms include Q-learning (Watkins, 1992), Deep Q Networks (DQN) (Mnih et al., 2013), and various actor-critic methods that utilize importance sampling or experience replay (Sutton & Barto, 2018). The principles of off-policy learning have also recently found fertile ground in generative modeling, notably in the framework of Generative Flow Networks (GFlowNets) (Bengio et al., 2021; Maddison et al., 2021). A key aspect of GFlowNets is their ability to train off-policy, learning from trajectories generated by any valid exploration policy. This off-policy capability is crucial for GFlowNets to effectively explore and cover the entire target distribution, especially in multimodal settings (Bengio et al., 2021; Lahlou et al., 2023). Other related efforts have investigated different learning objectives for GFlowNets that support off-policy training, such as the trajectory balance objective (Malkin et al., 2022) and subtrajectory balance (Madan et al., 2022), and explored their application to continuous spaces and diffusion models (Malkin et al., 2023; Zhang et al., 2024; Sendera et al., 2024).

## B  ALGORITHM

We list the detailed training algorithm of the proposed method in Algorithm 1 [1].

---

**Algorithm 1** Importance Weighted Score Matching

---

**Require:** Score network $s_\theta(x,t)$; energy function $E(x)$; VE-SDE parameters $\sigma_t$, prior $p_1$; Replay buffer $\mathcal{B}$; batch size for initial samples $B$ and SNIS samples $S$; inner loop steps $N_{\text{inner}}$. sample count $K$.

1: **for** outer loop iteration $= 1, 2, \ldots$ **do**
2:      Simulate reverse SDE (Eq. 3) using $s_\theta$ and fill $\mathcal{B}$.
3:      **for** inner loop step $= 1$ to $N_{\text{inner}}$ **do**
4:          Sample $\{x_0^{(b)}\}_{b=1}^B \sim \mathcal{B}$, $\{t_b\}_{b=1}^B \sim \text{Uni}[0,1]$.
5:          Sample $\{\{x_{t|0}^{(b,s)}\}_{s=1}^S \sim p(\cdot|x_0^{(b)})\}_{b=1}^B$, $\{\{\{x_{0|t}^{(b,s,k)}\}_{k=1}^K \sim \mathcal{N}(x_0; x_t^{(b,s)}, \sigma_t^2 I)\}_{s=1}^S\}_{b=1}^B$.
6:          Compute $\{\{\{E(x_{0|t}^{(b,s,k)})\}_{k=1}^K\}_{s=1}^S\}_{b=1}^B$, score target $\{\{S_L^{(b,s)}\}_{s=1}^S\}_{b=1}^B$ by Eq. 7.
7:          Compute SNIS weights $\{\{\tilde{w}_{\text{SNIS}}(x_{t|0}^{(b,s)})\}_{s=1}^S\}_{b=1}^B$ by Eq. 12.
8:          Compute $x_0$ conditioned loss $\{L_{\text{SNIS}}(\theta, t_b|x_0^{(b)})\}_{b=1}^B$ by Eq. 15.
9:          Update $s_\theta$ by loss back propagation on $L_{\text{batch}}(\theta, t)$ (Eq. 16).
10:      **end for**
11: **end for**

---

**Practical Considerations.** The computation of the batch loss $L_{\text{batch}}$ involves several Monte Carlo estimations which could introduce significant computational overhead due to auxiliary sampling requirements and energy evaluations. However, considerable efficiency is achieved through strategic sample reuse across the different estimation components. Firstly, the estimation of the weight numerator $N_K(x_t, t)$ (Eq. 9) and the score target $S_L(x_t, t)$ (Eq. 7) both rely on sampling from the identical proposal distribution ($\mathcal{N}(x_0; x_t, \sigma_t^2 I)$ in the context of VE-SDE) and subsequent energy evaluations $E(x_0)$. Consequently, by setting the number of samples identically, $K = L$, the same set of $K$ samples can be utilized for both calculations, reusing the energy evaluations. Secondly, the estimation of the weight denominator $D_M(x_t, t)$ (Eq. 10) requires $M$ samples $\{x_0^{(j)}\}$ from the buffer distribution $p_0^{\mathcal{B}}$ to evaluate the forward kernel $p_{t|0}(x_t|x_0^{(j)})$. A computationally efficient strategy involves utilizing the samples already loaded in the current mini-batch $\{x_0^{(i)}\}_{i=1}^B \sim \mathcal{B}$, then the need for auxiliary buffer sampling specifically for the denominator estimation is eliminated. In fact, the estimation of $D_M(x_t, t)$ corresponds to kernel density estimation using Gaussian kernels with bandwidth $\sigma_t$. At small noise levels, these Gaussian kernels approach Dirac distributions, which can lead to numerical instability. Our strategy of reusing mini-batch samples helps avoid such instabilities.

## C  DETAILS ON SCORE-BASED DIFFUSION MODELS

This appendix provides further details on the specific forms of Stochastic Differential Equations (SDEs) commonly used in score-based diffusion models, as mentioned in the main text. We focus on two prominent parameterizations: Variance Preserving SDE (VP-SDE) and Variance Exploding SDE (VE-SDE).

### C.1  GENERAL FORWARD AND REVERSE SDES

As described in the main text, a forward diffusion process transforms a data sample $x_0 \sim \pi(x)$ into a noisy sample $x_t$ over a time interval $t \in [0, T]$ (often normalized to $[0, 1]$) via the SDE:

$$dx_t = f(x_t, t)dt + g(t)dw_t, \tag{17}$$

where $w_t$ is a standard Wiener process, $f(x_t, t)$ is the drift coefficient, and $g(t)$ is the diffusion coefficient. The distribution of $x_t$ at time $t$ is denoted by $p_t(x_t)$. The corresponding reverse-time

---

[1]We employ multi-index superscripts, such as $(b, s)$ in $x_{0|t}^{(b,s)}$, to track sample provenance through the nested loops of Algorithm 1. Index $b$ $(1, \ldots, B)$ identifies the initial sample from the replay buffer batch. Index $s$ $(1, \ldots, S)$ identifies samples derived from $x_0^{(b)}$ (e.g., $x_{t|0}^{(b,s)}$ via forward diffusion).

SDE, which transforms samples from the prior distribution $p_T(x_T)$ back towards the data distribution $\pi(x) \equiv p_0(x)$, is given by:

$$dx_t = \left[ f(x_t, t) - g(t)^2 \nabla_{x_t} \log p_t(x_t) \right] dt + g(t) d\bar{w}_t, \tag{18}$$

where $d\bar{w}_t$ is a standard Wiener process when time flows backwards from $T$ to 0. In practice, the true score $\nabla_{x_t} \log p_t(x_t)$ is unknown and is approximated by a learned score model $s_\theta(x_t, t)$, leading to the model reverse SDE:

$$d\hat{x}_t = \left[ f(\hat{x}_t, t) - g(t)^2 s_\theta(\hat{x}_t, t) \right] dt + g(t) d\bar{w}_t. \tag{19}$$

## C.2 VARIANCE PRESERVING SDE

The VP-SDE is designed such that the variance of the perturbed data $x_t$ is asymptotically preserved or bounded. It is closely related to the Denoising Diffusion Probabilistic Models (DDPM) (Ho et al., 2020). A common formulation for the VP-SDE is:

$$f(x_t, t) = -\frac{1}{2} \beta(t) x_t, \ g(t) = \sqrt{\beta(t)}, \tag{20}$$

where $\beta(t)$ is a positive, monotonically increasing noise schedule, typically ranging from $\beta_{\min} = \beta(0)$ to $\beta_{\max} = \beta(T)$. For instance, $\beta(t)$ can be linearly interpolated between $\beta_{\min}$ (e.g., $10^{-4}$) and $\beta_{\max}$ (e.g., 0.02) over $t \in [0, T]$.

The transition kernel $p_{t|0}(x_t|x_0) = \mathcal{N}(x_t; \alpha(t)x_0, \sigma(t)^2 I)$ for this SDE has parameters:

$$\alpha(t) = \exp\left( -\frac{1}{2} \int_0^t \beta(s) ds \right), \ \sigma(t)^2 = 1 - \exp\left( -\int_0^t \beta(s) ds \right) = 1 - \alpha(t)^2. \tag{21}$$

As $t \to T$, assuming $\int_0^T \beta(s) ds \to \infty$, $\alpha(T) \to 0$ and $\sigma(T)^2 \to 1$, so $p_T(x_T)$ approaches $\mathcal{N}(0, I)$ if $x_0$ has zero mean and unit variance on average. The score matching objective for VP-SDE often involves predicting the noise $\epsilon$ in $x_t = x_0 \alpha(t) + \sigma(t)\epsilon$, where $\epsilon \sim \mathcal{N}(0, I)$. The score is related to this noise by $\nabla_{x_t} \log p_t(x_t) = -\epsilon/\sigma(t)$.

The reverse SDE for VP-SDE using the learned score $s_\theta(x_t, t)$ becomes:

$$d\hat{x}_t = \left[ -\frac{1}{2} \beta(t) \hat{x}_t - \beta(t) s_\theta(\hat{x}_t, t) \right] dt + \sqrt{\beta(t)} d\bar{w}_t. \tag{22}$$

## C.3 VARIANCE EXPLODING SDE

The VE-SDE allows the variance of the perturbed data $x_t$ to grow (explode) over time. It is related to an earlier generation of score-based models by Song & Ermon (2019). A common formulation for the VE-SDE is:

$$f(x_t, t) = 0, \ g(t) = \sqrt{\frac{d[\sigma(t)^2]}{dt}} \tag{23}$$

where $\sigma(t)$ is a positive, monotonically increasing function representing the standard deviation of the noise added at time $t$. For instance, $\sigma(t)$ might follow a geometric progression, $\sigma(t) = \sigma_{\min}(\sigma_{\max}/\sigma_{\min})^{t/T}$, where $t \in [0, T]$, $\sigma_{\min}$ is small (e.g., 0.01) and $\sigma_{\max}$ is large (e.g., 50).

The transition kernel $p_{t|0}(x_t|x_0) = \mathcal{N}(x_t; x_0, \sigma(t)^2 I)$. Note that here, the mean of $x_t$ given $x_0$ is simply $x_0$. The score matching objective for VE-SDE aims to estimate $\nabla_{x_t} \log p_t(x_t)$. The specific form of the conditional distribution $q(x_0|x_t)$ (or $p_{0|t}(x_0|x_t)$) used in some estimation procedures (like the one for $S_L$ in our Proposition 2, if it's within a VE-SDE context) is often derived from Bayes' theorem:

$$p_{0|t}(x_0|x_t) \propto p_{t|0}(x_t|x_0) p_0(x_0) = \mathcal{N}(x_t; x_0, \sigma(t)^2 I) p_0(x_0). \tag{24}$$

Then the reverse SDE for VE-SDE using the learned score $s_\theta(x_t, t)$ becomes:

$$dx_t = -g(t)^2 s_\theta(x_t, t) dt + g(t) d\bar{w}_t. \tag{25}$$

## C.4 CONNECTION TO ORDINARY DIFFERENTIAL EQUATIONS (ODEs)

For any diffusion process defined by Eq. 17, there exists a corresponding deterministic process (an Ordinary Differential Equation, ODE) whose trajectories share the same marginal probability densities $p_t(x_t)$ (Song et al., 2021). This probability flow ODE is given by:

$$dx_t = \left[ f(x_t, t) - \frac{1}{2} g(t)^2 \nabla_{x_t} \log p_t(x_t) \right] dt. \tag{26}$$

Replacing $\nabla_{x_t} \log p_t(x_t)$ with $s_\theta(x_t, t)$ gives a generative ODE model. ODE samplers are often more efficient as they allow for larger step sizes and can utilize adaptive step size solvers. This appendix provides a brief overview. For a comprehensive study, readers are referred to Song et al. (2021) and Karras et al. (2022).

# D TARGET SCORE DERIVATION

## D.1 STANDARD DERIVATION

Under the VE-SDE, the marginal distribution $p_t$ is expressed as a convolution of the initial distribution $p_0$ with a Gaussian kernel: $p_t(x_t) = (p_0 * \mathcal{N}(0, \sigma_t^2))(x_t)$, where $*$ denotes the convolution operation.

We commence by expressing the score function $\nabla \log p_t(x_t)$ in terms of the convolution representation:

$$\nabla \log p_t(x_t) = \frac{\nabla (p_0 * \mathcal{N}(0, \sigma_t^2))(x_t)}{p_t(x_t)}.$$

Employing the differential property of convolutions, which states that the gradient of a convolution equals the convolution of one function with the gradient of the other, we obtain:

$$\nabla \log p_t(x_t) = \frac{((\nabla p_0) * \mathcal{N}(0, \sigma_t^2))(x_t)}{p_t(x_t)}.$$

This expression can be reformulated in terms of expectations. Let $x_{0|t}$ denote a random variable with conditional distribution $\mathcal{N}(x_t, \sigma_t^2)$. Then:

$$\nabla \log p_t(x_t) = \frac{\mathbb{E}_{x_{0|t} \sim \mathcal{N}(x_t, \sigma_t^2)}[\nabla p_0(x_{0|t})]}{\mathbb{E}_{x_{0|t} \sim \mathcal{N}(x_t, \sigma_t^2)}[p_0(x_{0|t})]}.$$

Given that $p_0$ follows a Boltzmann distribution, i.e., $p_0(x) = \frac{\exp(-E(x))}{\mathcal{Z}}$ where $E$ represents an energy function and $\mathcal{Z}$ is the normalization constant, we derive:

$$\nabla \log p_t(x_t) = \frac{\mathbb{E}_{x_{0|t} \sim \mathcal{N}(x_t, \sigma_t^2)}[\nabla \exp(-E(x_{0|t}))]}{\mathbb{E}_{x_{0|t} \sim \mathcal{N}(x_t, \sigma_t^2)}[\exp(-E(x_{0|t}))]}.$$

The normalization constant $\mathcal{Z}$ appears identically in both numerator and denominator, thereby canceling out in this formulation. From the practical implementation level, we use Monte Carlo estimation on the expectation:

$$\nabla \log p_t(x_t) \approx \frac{\frac{1}{L} \sum_{i=1}^{L} \nabla \exp(-E(x_{0|t}^{(i)}))}{\frac{1}{L} \sum_{i=1}^{L} \exp(-E(x_{0|t}^{(i)}))} \overset{(*)}{=} \nabla_{x_t} \log \sum_{i=1}^{L} \exp(-E(x_{0|t}^{(i)})),$$

where $\{x_{0|t}^{(i)}\}_{i=1}^{L} \sim \mathcal{N}(x_0; x_t, \sigma_t^2 I)$ and the last equation $\overset{(*)}{=}$ results from the VE-SDE setting and reparameterization trick.

Following the derivation from Akhound-Sadegh et al. (2024), we then demonstrate how a general stochastic differential equation (SDE) can be transformed into a VE-SDE through an appropriate change of variables. Consider a general SDE of the form: $dx_t = f(x_t, t) \, dt + g(t) \, dw_t$, where $f(x_t, t)$ represents the drift term, $g(t)$ denotes the diffusion coefficient, and $w_t$ is the standard Wiener process. In the case where $f(x_t, t) = -\beta(t)x_t$, we can introduce a time-dependent rescaling of $x_t$ to transform the SDE into a VE form. Define:

$$y_t := \alpha(t)x_t, \quad \text{where} \quad \alpha(t) := \exp\left( -\int_0^t \beta(s) \, ds \right).$$

Applying Itô's lemma to derive the dynamics of $y_t$, we obtain:

$$dy_t = [\alpha'(t) - \alpha(t)\beta(t)]x_t \, dt + g(t)\alpha(t) \, dw_t.$$

Since $\alpha'(t) = -\alpha(t)\beta(t)$ by definition, the drift term vanishes, yielding:

$$dy_t = g(t)\alpha(t) \, dw_t,$$

which is precisely a variance-exploding SDE with no drift term. This transformation preserves the initial condition $y_0 = x_0$ and generates marginal densities $\tilde{p}_t(y_t)$ from the initial distribution $\tilde{p}_0 = p_0$. Importantly, estimating the score function $\nabla \log \tilde{p}_t(y_t)$ is equivalent to estimating $\nabla \log p_t(x_t)$, as they are related by:

$$\nabla \log \tilde{p}_t(y_t) = \nabla \log p_t(x_t).$$

From a practical perspective, whether the neural network estimator takes $x_t$ or the rescaled $y_t$ as input is an implementation choice; we opt for $x_t$ as input to enhance numerical stability.

### D.2 TARGET SCORE IDENTITY TRANSFORMATION

We begin with the target score identity for the VE-SDE noising schedule.

**Proposition 4** (See Proposition 2.1 in TSI (Bortoli et al., 2024)). *For the noising schedule of VE-SDE: $x_t = x_0 + \sigma_t \epsilon$ where $\epsilon \sim \mathcal{N}(0, I)$, the following Target Score Identity holds:*

$$\nabla \log p_t(x_t) = \int \nabla \log p_0(x_0) p_{0|t}(x_0|x_t) dx_0. \tag{27}$$

To derive a practical estimator, we transform this identity using Bayes' rule and introduce a Gaussian proposal distribution:

$$\nabla \log p_t(x_t) = \int \nabla \log p_0(x_0) p_{0|t}(x_0|x_t) dx_0 \tag{28}$$

$$= \int \nabla \log p_0(x_0) \frac{p_{t|0}(x_t|x_0) p_0(x_0)}{p_t(x_t)} dx_0 \tag{29}$$

$$= \int \mathcal{N}(x_0; x_t, \sigma_t^2 I) \nabla \log p_0(x_0) \frac{p_{t|0}(x_t|x_0) p_0(x_0)}{p_t(x_t) \mathcal{N}(x_0; x_t, \sigma_t^2 I)} dx_0 \tag{30}$$

$$= \int \mathcal{N}(x_0; x_t, \sigma_t^2 I) \nabla \log p_0(x_0) \frac{p_0(x_0)}{p_t(x_t)} dx_0, \tag{31}$$

where in the last step we used the fact that $p_{t|0}(x_t|x_0) = \mathcal{N}(x_t; x_0, \sigma_t^2 I) = \mathcal{N}(x_0; x_t, \sigma_t^2 I)$.

Since $p_t(x_t)$ is independent of $x_0$, we can apply self-normalized importance sampling with proposal distribution $q(x_0) = \mathcal{N}(x_0; x_t, \sigma_t^2 I)$ and importance weights $w(x_0) = p_0(x_0)/p_t(x_t)$. Drawing samples $\{x_{0|t}^{(i)}\}_{i=1}^L \sim \mathcal{N}(x_0; x_t, \sigma_t^2 I)$, the self-normalized importance sampling estimator yields:

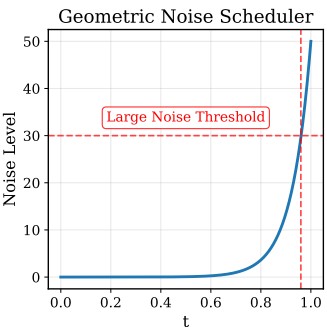

Geometric Noise Scheduler

Large Noise Threshold

Noise Level

$$\nabla \log p_t(x_t) \approx \frac{\sum_{i=1}^L w(x_{0|t}^{(i)}) \nabla \log p_0(x_{0|t}^{(i)})}{\sum_{i=1}^L w(x_{0|t}^{(i)})}$$

$$= \frac{\sum_{i=1}^L p_0(x_{0|t}^{(i)}) \nabla \log p_0(x_{0|t}^{(i)})}{\sum_{i=1}^L p_0(x_{0|t}^{(i)})}. \tag{32}$$

For energy-based models where $p_0(x) \propto \exp(-E(x))$, we have $\nabla \log p_0(x) = -\nabla E(x)$. After algebraic manipulation, this leads to our estimator in Eq. 7. The Gaussian proposal distribution $\mathcal{N}(x_0; x_t, \sigma_t^2 I)$ is particularly effective at small noise levels, as it concentrates around the denoised estimate and provides lower-variance estimates compared to other proposal choices.

Figure 4: Geometric noise schedule used in VE-SDE. Large noise occurs only for $t \geq 0.96$ approximately, representing less than 5% of the training time interval $[0, 1]$.

However, this estimator relies on importance sampling and thus suffers from high variance at large noise levels, potentially degrading estimation quality. Fortunately, our VE-SDE training employs a geometric noise schedule that concentrates most samples at smaller noise levels, as illustrated in Figure 4. This strategic design ensures accurate learning targets by maintaining low estimator variance in the regions where training samples are most frequently drawn, effectively circumventing the high-variance regime of importance sampling. Furthermore, Akhound-Sadegh et al. (2024) provides a comprehensive analysis of how the sample size $L$, noise level, and target distribution dimensionality affect estimator accuracy—we refer interested readers to Appendix G.3 therein for detailed empirical and theoretical insights.

## E  PROOF COLLECTIONS

### E.1  PROOF OF PROPOSITION 1

*Proof.* The proof utilizes Girsanov's theorem to relate the true reverse path measure $\mathbb{P}_r$ and the model reverse path measure $\mathbb{P}_\theta$.

We define the Radon-Nikodym derivative $\frac{\mathrm{d}\mathbb{P}_\theta}{\mathrm{d}\mathbb{P}_r}$ using Girsanov's theorem. Let the drift difference scaled by the diffusion be:

$$b(t, x) := g(t)\left(\nabla \log p_t(x) - s_\theta(x, t)\right).$$

Assume that $\mathbb{E}_{\mathbb{P}_r}\left[\int_0^1 \|b(s, x_s)\|^2 \mathrm{d}s\right] < +\infty$ (Novikov Condition), which, along with standard regularity conditions, ensures that the exponential martingale is a true martingale. Define the process $\mathcal{E}(\mathcal{L})_t$ for $t \in [0, 1]$ as

$$\mathcal{E}(\mathcal{L})_t := \exp\left(\int_0^t b(s, x_s)^\top \mathrm{d}\tilde{w}_s - \frac{1}{2}\int_0^t \|b(s, x_s)\|^2 \mathrm{d}s\right).$$

Then $\mathcal{E}(\mathcal{L})_t$ is a $\mathbb{P}_r$-martingale with $\mathbb{E}_{\mathbb{P}_r}[\mathcal{E}(\mathcal{L})_t] = 1$. Define the measure $\mathbb{P}_\theta$ on the path space by $\frac{\mathrm{d}\mathbb{P}_\theta}{\mathrm{d}\mathbb{P}_r} = \mathcal{E}(\mathcal{L})_1$.

By Girsanov's theorem, under the measure $\mathbb{P}_\theta$, the process

$$\beta_t := \tilde{w}_t - \int_0^t b(s, x_s)\mathrm{d}s$$

is a standard $\mathbb{P}_\theta$-Brownian motion. The dynamics of $x_t$ under $\mathbb{P}_r$ is:

$$\mathrm{d}x_t = \mu_r(t, x_t)\mathrm{d}t + g(t)\mathrm{d}\tilde{w}_t,$$

where $\mu_r = f - g^2 \nabla \log p_t$. Substituting $\mathrm{d}\tilde{w}_t = \mathrm{d}\beta_t + b(t, x_t)\mathrm{d}t$, the dynamics under $\mathbb{P}_\theta$ becomes:

$$\begin{aligned}
\mathrm{d}x_t &= \mu_r(t, x_t)\mathrm{d}t + g(t)(\mathrm{d}\beta_t + b(t, x_t)\mathrm{d}t) = (\mu_r(t, x_t) + g(t)b(t, x_t))\mathrm{d}t + g(t)\mathrm{d}\beta_t \\
&= \left((f(x_t, t) - g(t)^2 \nabla \log p_t(x_t)) + g(t) \cdot g(t)(\nabla \log p_t(x_t) - s_\theta(x_t, t))\right)\mathrm{d}t + g(t)\mathrm{d}\beta_t \\
&= \left(f(x_t, t) - g(t)^2 \nabla \log p_t(x_t) + g(t)^2 \nabla \log p_t(x_t) - g(t)^2 s_\theta(x_t, t)\right)\mathrm{d}t + g(t)\mathrm{d}\beta_t \\
&= \left(f(x_t, t) - g(t)^2 s_\theta(x_t, t)\right)\mathrm{d}t + g(t)\mathrm{d}\beta_t.
\end{aligned}$$

This derived dynamics under $\mathbb{P}_\theta$ matches the model reverse-time SDE (Eq. 3), confirming that $\mathbb{P}_\theta$ is indeed the model reverse path measure.

The KL divergence between $\mathbb{P}_r$ and $\mathbb{P}_\theta$ is given by

$$\mathrm{KL}(\mathbb{P}_r \parallel \mathbb{P}_\theta) = \mathbb{E}_{\mathbb{P}_r}\left[\ln \frac{\mathrm{d}\mathbb{P}_r}{\mathrm{d}\mathbb{P}_\theta}\right] = \mathbb{E}_{\mathbb{P}_r}\left[-\ln \mathcal{E}(\mathcal{L})_1\right].$$

Expanding $\ln \mathcal{E}(\mathcal{L})_1$, we have:

$$\ln \mathcal{E}(\mathcal{L})_1 = \int_0^1 b(t, x_t)^\top \mathrm{d}\tilde{w}_t - \frac{1}{2}\int_0^1 \|b(t, x_t)\|^2 \mathrm{d}t.$$

Taking the expectation under $\mathbb{P}_r$, the stochastic integral term vanishes because it is a $\mathbb{P}_r$-martingale with initial value 0:

$$\mathbb{E}_{\mathbb{P}_r}\left[\int_0^1 b(t, x_t)^\top \mathrm{d}\tilde{w}_t\right] = 0.$$

Thus, the KL divergence simplifies to:

$$\mathrm{KL}(\mathbb{P}_r \parallel \mathbb{P}_\theta) = \frac{1}{2}\mathbb{E}_{\mathbb{P}_r}\left[\int_0^1 \|b(t, x_t)\|^2 \mathrm{d}t\right].$$

Substituting the definition of $b(t, x)$:

$$\|b(t, x)\|^2 = \|g(t)(\nabla \log p_t(x) - s_\theta(x, t))\|^2 = g(t)^2 \|s_\theta(x, t) - \nabla \log p_t(x)\|^2.$$

Therefore, by swapping the expectation over paths and the time integral (justified by Fubini's theorem under the stated conditions) and noting that the expectation $\mathbb{E}_{\mathbb{P}_r}$ at time $t$ corresponds to averaging over $x_t \sim p_t$, and the integral over $[0, 1]$ corresponds to an expectation over $t \sim U[0, 1]$ multiplied by 1, we get:

$$\mathrm{KL}(\mathbb{P}_r \parallel \mathbb{P}_\theta) = \frac{1}{2}\int_0^1 \mathbb{E}_{\mathbb{P}_r}\left[g(t)^2 \|s_\theta(x_t, t) - \nabla \log p_t(x_t)\|^2\right]\mathrm{d}t$$

$$= \frac{1}{2}\int_0^1 \mathbb{E}_{x_t \sim p_t}\left[g(t)^2 \|s_\theta(x_t, t) - \nabla \log p_t(x_t)\|^2\right]\mathrm{d}t$$

$$= \mathbb{E}_{t \sim U[0,1]}\mathbb{E}_{x_t \sim p_t}\left[\frac{g(t)^2}{2}\|s_\theta(x_t, t) - \nabla \log p_t(x_t)\|^2\right].$$

This establishes the desired connection between the KL divergence of path measures and the score matching objective, up to an additive constant independent of $\theta$. $\qquad\square$

### E.2 Proof of Proposition 2

*Proof.* The score estimator is $S_L(x_t, t) = \nabla_{x_t} \log \sum_{i=1}^L \exp(-E(x_{0|t}^{(i)}))$, where $\{x_{0|t}^{(i)}\}_{i=1}^L$ are i.i.d. samples from $q(x_0|x_t) = \mathcal{N}(x_0; x_t, \sigma_t^2 I)$. Assuming $\sigma_t$ is independent of $x_t$, $x_{0|t}^{(i)} = x_t + \sigma_t \varepsilon^{(i)}$ with $\varepsilon^{(i)} \sim \mathcal{N}(0, I)$, so $\nabla_{x_t} x_{0|t}^{(i)} = I$. The estimator can be expressed as $S_L(x_t, t) = D_L/N_L$, where

$$N_L = N_L(x_t, t) = \frac{1}{L}\sum_{i=1}^L \exp(-E(x_{0|t}^{(i)})),$$

$$D_L = D_L(x_t, t) = \frac{1}{L}\sum_{i=1}^L \left[-\exp(-E(x_{0|t}^{(i)}))\nabla E(x_{0|t}^{(i)})\right].$$

The target score is $\nabla_{x_t} \log p_t(x_t) = \mu_D/\mu_N$, where $\mu_N = \mathbb{E}[N_L]$ and $\mu_D = \mathbb{E}[D_L]$, with expectations over $x_{0|t}^{(i)} \sim q(x_0|x_t)$. We adopt the notation $\mu_N(x_t, t) \equiv \exp(-E_t(x_t))$ and $\mu_D(x_t, t) \equiv \nabla_{x_t} \exp(-E_t(x_t))$.

The assumptions that $E(x) \geq E_{\min}$ and $\|\nabla E(x)\| \leq M_E$ imply that the random variables $Y^{(i)} = \exp(-E(x_{0|t}^{(i)}))$ and the random vectors $Z^{(i)} = -\exp(-E(x_{0|t}^{(i)}))\nabla E(x_{0|t}^{(i)})$ are bounded. Specifically, $0 < Y^{(i)} \leq \exp(-E_{\min})$ and $\|Z^{(i)}\| \leq \exp(-E_{\min})M_E$. Furthermore, $\exp(-E(x))$ and $|\nabla \exp(-E(x))|$ are Lipschitz functions, which leads to $\exp(-E(x_{0|t}^{(i)}))$ and $|\nabla \exp(-E(x_{0|t}^{(i)}))|$ are sub-Gaussian variables. (Wainwright, 2019, Theorem 2.26)

By Hoeffding's inequality, for any $\delta > 0$, there exist constants $K_N, K_D > 0$ such that with probability at least $1 - \delta$:

$$|N_L - \mu_N| \leq K_N\sqrt{\frac{\log(2/\delta)}{L}}, \quad \|D_L - \mu_D\| \leq K_D\sqrt{\frac{\log(2/\delta)}{L}}.$$

Let $K = \max(K_N, K_D)$. The error term is, for som large $L$ (e.g., $L \geq 4K_N^2 \log(2/\delta)/\mu_N^2$),

$$\left\| S_L - \frac{\mu_D}{\mu_N} \right\| = \left\| \frac{D_L N_L^{-1} \mu_N - \mu_D}{\mu_N} \right\| = \left\| \frac{(D_L - \mu_D)\mu_N - (N_L - \mu_N)\mu_D}{N_L \mu_N} \right\|$$

$$\leq \frac{\|(D_L - \mu_D)\mu_N\| + \|(N_L - \mu_N)\mu_D\|}{|N_L||\mu_N|} \leq \frac{K\sqrt{\frac{\log(2/\delta)}{L}}(|\mu_N| + \|\mu_D\|)}{|N_L||\mu_N|}, \quad \text{w.p. } \geq 1 - \delta.$$

Since $\mu_N > 0$, for $L$ sufficiently large such that $K_N\sqrt{\log(2/\delta)/L} \leq \mu_N/2$, we have $|N_L| \geq \mu_N/2$. Thus, with probability at least $1 - \delta$:

$$\|S_L(x_t, t) - \nabla_{x_t} \log p_t(x_t)\| \leq \frac{2K(|\mu_N| + \|\mu_D\|)}{\mu_N^2}\sqrt{\frac{\log(2/\delta)}{L}} =: C_1(x_t, t)\sqrt{\frac{\log(2/\delta)}{L}},$$

where term $C_1(x_t, t)$ depends on $x_t, t$ and $E_{\min}, M_E$. This implies that $\|S_L(x_t, t) - \nabla_{x_t} \log p_t(x_t)\|$ has a sub-Gaussian tail:

$$\mathbb{P}\left(\|S_L(x_t, t) - \nabla_{x_t} \log p_t(x_t)\| > \epsilon\right) \leq 2\exp\left(-\frac{\epsilon^2}{2(M_E \cdot e^{-\min})^2}\right) \quad \text{for } \epsilon > 0.$$

For a random variable $X$ with $\mathbb{P}(|X| > \epsilon) \leq 2\exp(-\epsilon^2/(2\sigma^2))$, it holds that $\mathbb{E}[|X|] \leq K_0 \sigma$ for some universal constant $K_0$ (e.g., $K_0 = \sqrt{2\pi}$) (Wainwright, 2019, Chapter 2). Thus, applying Jensen's inequality:

$$\|\mathbb{E}[S_L(x_t, t)] - \nabla_{x_t} \log p_t(x_t)\| \leq \mathbb{E}[\|S_L(x_t, t) - \nabla_{x_t} \log p_t(x_t)\|]$$

$$\leq \frac{C_1(x_t, t)K_0}{\sqrt{2L}} =: \frac{c'(x_t, t)}{\sqrt{L}}. \quad (*)$$

For the variance, we use the property $\text{Var}(X) = \mathbb{E}[\|X - \mathbb{E}[X]\|^2] \leq \mathbb{E}[\|X - y\|^2]$ for any fixed $y$.

$$\text{Var}[S_L(x_t, t)] \leq \mathbb{E}[\|S_L(x_t, t) - \nabla_{x_t} \log p_t(x_t)\|^2].$$

The squared norm of a sub-Gaussian random vector also exhibits concentration, and its expectation is bounded by its variance proxy. Specifically, if $\|X\|$ is $\sigma^2$-sub-Gaussian, $\mathbb{E}[\|X\|^2] \leq K_1 \sigma^2$ for some constant $K_1$ (Wainwright, 2019, Chapter 2). Here, the variance proxy is $C_1(x_t, t)^2/(2L)$. Thus,

$$\mathbb{E}[\|S_L(x_t, t) - \nabla_{x_t} \log p_t(x_t)\|^2] \leq \frac{C_1(x_t, t)^2 K_1}{2L} =: \frac{c''(x_t, t)^2}{L}. \quad (**)$$

We denote $c(x_t, t)$ as $\max\{c'(x_t, t), c''(x_t, t)\}$, then the proposition follows by defining $c(x_t, t)$ appropriately based on $(*)$ and $(**)$. $\qquad\square$

### E.3 Proof of Proposition 3

*Proof.* Let $p_t(\cdot)$ denote the target distribution and $p_t^{\mathcal{B}}(\cdot)$ the proposal distribution for $x_t$. We assume $p_t \ll p_t^{\mathcal{B}}$. The unnormalized importance weight estimate $\tilde{w}(x_t)$ is defined as

$$\tilde{w}(x_t) = \frac{N_K(x_t)}{D_M(x_t)} = \frac{\frac{1}{K}\sum_{k=1}^{K} \exp(-E(x_{0|t}^{(k)}))}{\frac{1}{M}\sum_{j=1}^{M} p_{t|0}(x_t|x_0^{(j)})},$$

where $x_{0|t}^{(k)} \sim \mathcal{N}(x_0; x_t, \sigma_t^2 I)$ are i.i.d. samples for the numerator, and $x_0^{(j)} \sim p_0^{\mathcal{B}}(x_0)$ are i.i.d. samples for the denominator, with $p_{t|0}(\cdot|\cdot)$ being the VE-SDE forward transition kernel. This $\tilde{w}(x_t)$ serves as an estimate for $Z \cdot p_t(x_t)/p_t^{\mathcal{B}}(x_t)$ for some normalization constant $Z$. The Self-Normalized Importance Sampling (SNIS) estimator for $\mathbb{E}_{x_t \sim p_t}[f(x_t)]$ is

$$\hat{L}_{\text{SNIS}}[f] = \sum_{s=1}^{S} \frac{\tilde{w}(x_t^{(s)})}{\sum_{j'=1}^{S} \tilde{w}(x_t^{(j')})} f(x_t^{(s)}), \quad \{x_t^{(s)}\}_{s=1}^{S} \stackrel{\text{i.i.d.}}{\sim} p_t^{\mathcal{B}}.$$

Let $l(x_t, t) = \|s_\theta(x_t, t) - S_L(x_t, t)\|^2$ be the inner loss, where $S_L(x_t, t)$ is the score estimator from Proposition 2. The estimator under consideration is $L_{\text{SNIS}}(\theta, t) = \hat{L}_{\text{SNIS}}[l]$. The target loss is

$L^*(\theta, t) = \mathbb{E}_{x_t \sim p_t}[l^*(x_t, t)]$, where $l^*(x_t, t) = \|s_\theta(x_t, t) - \nabla_{x_t} \log p_t(x_t)\|^2$. Let $\mathbb{E}[\cdot]$ denote the total expectation over all sources of randomness: the batch $\mathcal{S} = \{x_t^{(s)}\}_{s=1}^S$, the internal samples $\{x_{0|t}^{(i)}\}_{i=1}^L$ for each $S_L(x_t^{(s)}, t)$ (denoted $\mathbb{E}_{S_L}$), and the internal samples for each $\tilde{w}(x_t^{(s)})$.

The bias is decomposed as:

$$\mathbb{E}[L_{\text{SNIS}}(\theta, t)] - L^*(\theta, t) = \underbrace{(\mathbb{E}[L_{\text{SNIS}}(\theta, t)] - \mathbb{E}_{x_t \sim p_t}[\mathbb{E}_{S_L}[l(x_t, t)]])}_{I_1 \text{ (SNIS bias for } \mathbb{E}_{S_L}[l])}$$

$$+ \underbrace{(\mathbb{E}_{x_t \sim p_t}[\mathbb{E}_{S_L}[l(x_t, t)]] - \mathbb{E}_{x_t \sim p_t}[l^*(x_t, t)])}_{I_2 \text{ (Bias from } S_L \text{ approx. to } \nabla \log p_t)}.$$

To facilitate the subsequent analysis, we begin by laying out the fundamental assumptions that underpin our results.

**Assumptions:**

A1. The inner loss, after averaging over $S_L$'s randomness, is uniformly bounded by $C_1$: $\mathbb{E}_{S_L}[\|s_\theta(x_t, t) - S_L(x_t, t)\|^2] \leq C_1$ for all $x_t, t$.

A2. The parameterized model error is uniformly bounded: $\|s_\theta(x_t, t) - \nabla_{x_t} \log p_t(x_t)\| \leq C_2$ for all $x_t, t$.

A3. Let $\mu_{\tilde{w}} = \mathbb{E}_{x_t \sim p_t^{\mathcal{B}}}[\mathbb{E}_{\text{samples of } \tilde{w}}[\tilde{w}(x_t)]]$, $V_{\tilde{w}} := \frac{\mathbb{E}_{x_t \sim p_t^{\mathcal{B}}}[\mathbb{E}_{\text{samples of } \tilde{w}}[\tilde{w}(x_t)^2]]}{\mu_{\tilde{w}}^2} < \infty$ and assume $\mu_{\tilde{w}} \neq 0$.

A4. From Proposition 2, let $c_{S_L,1}(x_t, t)$ and $c_{S_L,2}(x_t, t)$ be such that:

- $\mathbb{E}_{S_L}[\|S_L(x_t, t) - \nabla_{x_t} \log p_t(x_t)\|] \leq \frac{c_{S_L}(x_t, t)}{\sqrt{L}}$.

- $\mathbb{E}_{S_L}[\|S_L(x_t, t) - \nabla_{x_t} \log p_t(x_t)\|^2] \leq \frac{c_{S_L}(x_t, t)^2}{L}$.

Define $\bar{c}_1 := \mathbb{E}_{x_t \sim p_t}[c_{S_L}(x_t, t)]$ and $\bar{c}_2 := \mathbb{E}_{x_t \sim p_t}[c_{S_L}(x_t, t)^2]$.

**On the Mildness of Assumptions.** These assumptions are generally mild or standard in the context of analyzing learning algorithms. Assumptions A1 and A2 posit uniform bounds on key error terms. A2 implies that the parameterized model $s_\theta$ does not deviate arbitrarily far from the true score, a common objective in score-based modeling. A1 bounds the expected squared difference between $s_\theta$ and the score estimator $S_L$, which can be reasonable if both $s_\theta$ and $S_L$ operate within certain limits, potentially ensured by network architecture or training regularization. A3 requires the first moment of the weight estimates to be non-zero and the second moment to be finite. The condition $\mu_{\tilde{w}} \neq 0$ is a technical requirement for self-normalized importance sampling, ensuring that the proposal distribution $p_t^{\mathcal{B}}$ has some overlap with the target $p_t$ as captured by $\tilde{w}$; if $\mu_{\tilde{w}} = 0$, it would imply a complete mismatch where $L_{\text{SNIS}}$ is ill-defined. Finite $V_{\tilde{w}}$ is crucial for controlling the variance of importance sampling estimators and is a standard assumption. Finally, A4 directly follows from the conclusions of Proposition 2, provided that the functions $c_{S_L,1}(x_t, t)$ and $c_{S_L,2}(x_t, t)^2$ are integrable with respect to $p_t(x_t)$.

We are now equipped to derive the bounds for the bias and MSE. The proof proceeds by analyzing the components $I_1$ and $I_2$ separately.

*Bounding $I_1$:* Under assumptions A1 and A3, and assuming Theorem 2.1 of Agapiou et al. (2017) applies to the estimated weights $\tilde{w}(x_t)$ using the function $\mathbb{E}_{S_L}[l(x_t, t)]$ (which is bounded by $C_1$), the bias of $L_{\text{SNIS}}(\theta, t)$ for estimating $\mathbb{E}_{x_t \sim p_t}[\mathbb{E}_{S_L}[l(x_t, t)]]$ is bounded by:

$$|I_1| \leq \frac{12 C_1 V_{\tilde{w}}}{S}.$$

*Bounding $I_2$ :* The second term, $I_2$, represents the bias introduced by using the score estimator $S_L(x_t, t)$ as a proxy for the true score $\nabla_{x_t} \log p_t(x_t)$ within the expected inner loss. Specifically,

$$I_2 = \mathbb{E}_{x_t \sim p_t} \left[ \mathbb{E}_{S_L} \left[ \|s_\theta(x_t, t) - S_L(x_t, t)\|^2 - \|s_\theta(x_t, t) - \nabla_{x_t} \log p_t(x_t)\|^2 \right] \right].$$

$$|I_2| \leq \mathbb{E}_{x_t \sim p_t} \left[ \mathbb{E}_{S_L} \left[ |\|s_\theta - S_L\|^2 - \|s_\theta - \nabla \log p_t\|^2| \right] \right]$$

$$= \mathbb{E}_{x_t \sim p_t} \left[ \mathbb{E}_{S_L} \left[ |(\|s_\theta - S_L\| - \|s_\theta - \nabla \log p_t\|)(\|s_\theta - S_L\| + \|s_\theta - \nabla \log p_t\|)| \right] \right].$$

Using the triangle inequality $\|s_\theta - S_L\| \leq \|s_\theta - \nabla \log p_t\| + \|S_L - \nabla \log p_t\|$:

$$|I_2| \leq \mathbb{E}_{x_t \sim p_t} \left[ \mathbb{E}_{S_L} \left[ \|S_L - \nabla \log p_t\| (\|S_L - \nabla \log p_t\| + 2\|s_\theta - \nabla \log p_t\|) \right] \right]$$

$$= \mathbb{E}_{x_t \sim p_t} \left[ \mathbb{E}_{S_L}[\|S_L - \nabla \log p_t\|^2] + 2\mathbb{E}_{S_L}[\|S_L - \nabla \log p_t\|] \cdot \|s_\theta - \nabla \log p_t\| \right].$$

Using assumptions A2 (for $C_2$) and A4 (for $c_{S_L,1}, c_{S_L,2}$):

$$|I_2| \leq \mathbb{E}_{x_t \sim p_t} \left[ \frac{c_{S_L}(x_t, t)^2}{L} + 2\frac{c_{S_L}(x_t, t)}{\sqrt{L}} C_2 \right]$$

$$= \frac{\mathbb{E}_{x_t \sim p_t}[c_{S_L}(x_t, t)^2]}{L} + \frac{2C_2 \mathbb{E}_{x_t \sim p_t}[c_{S_L}(x_t, t)]}{\sqrt{L}} \tag{33}$$

$$= \frac{\bar{c}_2}{L} + \frac{2C_2 \bar{c}_1}{\sqrt{L}} =: B_{S_L}.$$

Combining the bounds for $|I_1|$ and $|I_2|$, the total bias of $L_{\text{SNIS}}(\theta, t)$ with respect to $L^*(\theta, t)$ is bounded as:

$$|\mathbb{E}[L_{\text{SNIS}}(\theta, t)] - L^*(\theta, t)| \leq |I_1| + |I_2| \leq \frac{12C_1 V_{\tilde{w}}}{S} + \frac{\bar{c}_2}{L} + \frac{2C_2 \bar{c}_1}{\sqrt{L}}.$$

This completes the proof for the bias bound.

*Mean Squared Error:* We now turn to bounding the Mean Squared Error (MSE) of $L_{\text{SNIS}}(\theta, t)$ with respect to the target loss $L^*(\theta, t)$. Recall our decomposition where

$$L_{\text{SNIS}}(\theta, t) - L^*(\theta, t) = (L_{\text{SNIS}}(\theta, t) - \mathbb{E}_{p_t}[\mathbb{E}_{S_L}[l]]) + (\mathbb{E}_{p_t}[\mathbb{E}_{S_L}[l]] - L^*).$$

Let $X_{\text{SNIS}} := L_{\text{SNIS}}(\theta, t) - \mathbb{E}_{x_t \sim p_t}[\mathbb{E}_{S_L}[l(x_t, t)]]$ represent the error of the SNIS estimator with respect to its direct target $\mathbb{E}_{p_t}[\mathbb{E}_{S_L}[l]]$. The second term, $I_2 := \mathbb{E}_{x_t \sim p_t}[\mathbb{E}_{S_L}[l(x_t, t)]] - L^*(\theta, t)$, is the bias component analyzed previously, which is a deterministic quantity once all expectations are taken, bounded by $|I_2| \leq B_{S_L}$. The MSE can thus be expanded as:

$$\mathbb{E}[(L_{\text{SNIS}}(\theta, t) - L^*(\theta, t))^2] = \mathbb{E}[(X_{\text{SNIS}} + I_2)^2]$$

$$= \mathbb{E}[X_{\text{SNIS}}^2] + 2 \cdot \mathbb{E}[X_{\text{SNIS}}] \cdot I_2 + I_2^2. \tag{34}$$

We will bound each term on the right-hand side of Eq. 34.

The first term, $\mathbb{E}[X_{\text{SNIS}}^2]$, is the MSE of the SNIS estimator $L_{\text{SNIS}}(\theta, t)$ for estimating $\mathbb{E}_{x_t \sim p_t}[\mathbb{E}_{S_L}[l(x_t, t)]]$. Under assumptions A1 (which states that $\mathbb{E}_{S_L}[l(x_t, t)] \leq C_1$) and A3 (regarding weight moments), and by applying Theorem 2.1 of Agapiou et al. (2017), this term is bounded by:

$$\mathbb{E}[X_{\text{SNIS}}^2] \leq \frac{4C_1^2 V_{\tilde{w}}}{S}.$$

The second term involves the expectation $\mathbb{E}[X_{\text{SNIS}}]$, which is precisely the bias of $L_{\text{SNIS}}(\theta, t)$ with respect to $\mathbb{E}_{x_t \sim p_t}[\mathbb{E}_{S_L}[l(x_t, t)]]$. This is the quantity $I_1$ we bounded earlier ($|I_1| \leq \frac{12C_1 V_{\tilde{w}}}{S}$). Therefore, the absolute value of the cross-term can be bounded as:

$$|2 \cdot \mathbb{E}[X_{\text{SNIS}}] \cdot I_2| = |2 \cdot I_1 \cdot I_2| \leq 2|I_1||I_2| \leq 2\left(\frac{12C_1 V_{\tilde{w}}}{S}\right)\left(\frac{\bar{c}_2}{L} + \frac{2C_2 \bar{c}_1}{\sqrt{L}}\right).$$

The third term is simply the square of the bound for $|I_2|$:

$$I_2^2 \leq \left(\frac{\bar{c}_2}{L} + \frac{2C_2 \bar{c}_1}{\sqrt{L}}\right)^2.$$

Summing the bounds for these three terms from Eq. 34, we arrive at the overall MSE bound:

$$\mathbb{E}[(L_{\text{SNIS}}(\theta,t) - L^*(\theta,t))^2] \le \frac{4C_1^2 V_{\tilde{w}}}{S} + 2\left(\frac{12 C_1 V_{\tilde{w}}}{S}\right)\left(\frac{\bar{c}_2}{L} + \frac{2C_2 \bar{c}_1}{\sqrt{L}}\right) + \left(\frac{\bar{c}_2}{L} + \frac{2C_2 \bar{c}_1}{\sqrt{L}}\right)^2.$$

This expression can be compactly written by letting $B_{S_L} := \frac{\bar{c}_2}{L} + \frac{2C_2 \bar{c}_1}{\sqrt{L}}$ as defined in Eq. 33:

$$\text{MSE} \le \frac{4C_1^2 V_{\tilde{w}}}{S} + B_{S_L}\left[\frac{24 C_1 V_{\tilde{w}}}{S} + B_{S_L}\right].$$

This completes the proof for the MSE bound.

Finally, setting $K_1 = 12C_1$, $K_2 = 2C_2 \bar{c}_1$, $K_3 = \bar{c}_2$, we establish the desired results stated in the main text. $\qquad\square$

## F  RELATIONSHIPS BETWEEN DIFFERENT TRAINING OBJECTIVES

We clarify the relationships between our importance-weighted objective, iDEM's approach, and standard reverse KL divergence minimization. The key distinction lies in the replay buffer composition and its effect on the training objective.

The standard reverse KL approach assumes samples are drawn from the current model distribution $p_t^\theta$, optimizing

$$L_{\text{RKL}}(\theta) = \mathbb{E}_{t \sim U(0,1)} \mathbb{E}_{x_t \sim p_t^\theta}\left[\|s_\theta(x_t, t) - \nabla \log p_t(x_t)\|^2\right].$$

This on-policy training requires the buffer to contain only samples from the current model, necessitating frequent resampling and making the process computationally expensive.

In contrast, iDEM (Akhound-Sadegh et al., 2024) adopts an off-policy approach by maintaining a buffer with historical samples from previous model iterations, leading to the objective

$$L_{\text{iDEM}}(\theta) = \mathbb{E}_{t \sim U(0,1)} \mathbb{E}_{x_t \sim p_t^{\mathcal{B}}}\left[\|s_\theta(x_t, t) - \nabla \log p_t(x_t)\|^2\right],$$

where $p_t^{\mathcal{B}}$ mixes samples from different iterations. This historical mixing smooths the proposal distribution and mitigates mode collapse, producing more "mass-covering" samples. However, it effectively optimizes $D_{\text{Fisher}}(p_t^{\mathcal{B}} \| p_t^\theta)$ rather than the intended $D_{\text{Fisher}}(p_t \| p_t^\theta)$, leading to biased gradients and potentially noisier samples when $p_t^{\mathcal{B}}$ deviates significantly from $p_t$.

Our method builds upon iDEM's off-policy framework but introduces importance weighting to correct for the distribution mismatch:

$$L_{\text{ours}}(\theta) = \mathbb{E}_{t \sim U(0,1)} \mathbb{E}_{x_t \sim p_t^{\mathcal{B}}}\left[\frac{p_t(x_t)}{p_t^{\mathcal{B}}(x_t)}\|s_\theta(x_t, t) - \nabla \log p_t(x_t)\|^2\right].$$

This correction ensures unbiased gradient estimates while preserving the computational efficiency and exploration benefits of historical buffering. By accounting for the distribution mismatch between $p_t^{\mathcal{B}}$ and $p_t$, our method combines iDEM's practical advantages with theoretical convergence guarantees, particularly crucial for challenging multi-modal distributions.

## G  EXPERIMENT DETAILS

### G.1  BENCHMARK DISTRIBUTIONS

To rigorously evaluate our proposed methodology, we employ several canonical probability distributions that have been established in the literature for benchmarking generative models. These distributions exhibit varying characteristics that test different aspects of our algorithm's performance.

**Multivariate Gaussian Mixture Model.** For our first benchmark, we consider a two-dimensional Gaussian mixture model comprising $m \in \{40, 80, 120\}$ components with equiprobable weights. This distribution is formulated as:

$$p_{\text{GMM}}(\mathbf{x}) = \frac{1}{m} \sum_{i=1}^{m} \mathcal{N}(\mathbf{x}; \boldsymbol{\mu}_i, \boldsymbol{\Sigma}) \tag{35}$$

where each component is characterized by a unique mean vector $\boldsymbol{\mu}_i$ and a shared covariance matrix $\boldsymbol{\Sigma}$. The covariance structure is specified as:

$$\boldsymbol{\Sigma} = \begin{pmatrix} m & 0 \\ 0 & m \end{pmatrix}. \tag{36}$$

The mean vectors $\{\boldsymbol{\mu}_i\}_{i=1}^m$ are sampled from a uniform distribution over a square region: $\boldsymbol{\mu}_i \sim \mathcal{U}(-m, m)^2$. For consistency with established protocols in the literature Midgley et al. (2023b); Akhound-Sadegh et al. (2024), we utilize a test dataset comprising 10,000 samples generated using a predetermined random seed.

**Double-Well Four-Particle System (DW-4).** Our second benchmark concerns a physical system consisting of four particles interacting in a two-dimensional space through a double-well potential (Köhler et al., 2020). The configuration of this system is represented by a vector $\mathbf{x} = \{\mathbf{x}_1, \mathbf{x}_2, \mathbf{x}_3, \mathbf{x}_4\}$, where each $\mathbf{x}_i \in \mathbb{R}^2$ denotes the position of the $i$-th particle.

The energy function governing this system is defined as:

$$E^{\text{DW}}(\mathbf{x}) = \frac{1}{2\tau} \sum_{i<j} \left[ a(d_{ij} - d_0) + b(d_{ij} - d_0)^2 + c(d_{ij} - d_0)^4 \right] \tag{37}$$

where $d_{ij} = \|\mathbf{x}_i - \mathbf{x}_j\|_2$ represents the Euclidean distance between particles $i$ and $j$. Following conventions established in prior research, we parameterize this energy function with $a = 0$, $b = -4$, $c = 0.9$, and temperature parameter $\tau = 1$.

The probability density associated with this system follows the Boltzmann distribution:

$$p(\mathbf{x}) \propto \exp\left(-E^{\text{DW}}(\mathbf{x})\right). \tag{38}$$

For empirical validation, we utilize 10,000 samples from Markov Chain Monte Carlo simulations as reference data, acknowledging the inherent limitations of such approximations for ground truth assessment (Klein et al., 2023).

**Lennard-Jones Systems** The Lennard-Jones potential (Köhler et al., 2020) represents a fundamental model in molecular dynamics that captures both attractive and repulsive interactions between particles. For a system of $N$ particles, each with position $\mathbf{x}_i \in \mathbb{R}^3$, the energy function is expressed as:

$$E^{\text{LJ}}(\mathbf{x}) = \frac{\epsilon}{2\tau} \sum_{i<j} \left[ \left(\frac{r_m}{d_{ij}}\right)^{12} - \left(\frac{r_m}{d_{ij}}\right)^6 \right] \tag{39}$$

where $d_{ij} = \|\mathbf{x}_i - \mathbf{x}_j\|_2$ denotes the interparticle distance, and $r_m$, $\tau$, and $\epsilon$ are physical constants determining the characteristic distance, temperature, and energy scale, respectively.

To ensure spatial localization of the particle ensemble, we augment the Lennard-Jones potential with a harmonic oscillator term:

$$E^{\text{osc}}(\mathbf{x}) = \frac{1}{2} \sum_{i=1}^{N} \|\mathbf{x}_i - \mathbf{x}_{\text{COM}}\|^2 \tag{40}$$

where $\mathbf{x}_{\text{COM}}$ represents the center of mass of the system. The composite energy function is then defined as:

$$E^{\text{Total}}(\mathbf{x}) = E^{\text{LJ}}(\mathbf{x}) + c \cdot E^{\text{osc}}(\mathbf{x}) \tag{41}$$

with $c$ controlling the relative strength of the oscillator potential.

We examine two instantiations of this system:

- LJ-13: A system of 13 particles in three-dimensional space, resulting in a 39-dimensional configuration space.
- LJ-55: A system of 55 particles in three-dimensional space, yielding a high-dimensional problem with 165 degrees of freedom.

The Lennard-Jones systems present particular challenges for sampling algorithms due to the singularity in the energy function as any $d_{ij} \to 0$, leading to exploding score values in regions of the configuration space where particles approach one another.

For all Lennard-Jones experiments, we adopt the parameter values $r_m = 1$, $\tau = 1$, $\epsilon = 1$, and $c = 0.5$, consistent with established protocols in the literature. Evaluation is conducted using 10,000 reference samples generated through extensive Markov Chain Monte Carlo simulations (Klein et al., 2023).

### G.2 Details on Evaluation Metrics

We employ a diverse set of metrics to evaluate different aspects of the generated distributions, ranging from global distributional similarity to specific structural characteristics.

**Wasserstein Distances.** The Wasserstein distance, also known as the Earth Mover's Distance (Villani & Villani, 2009; Panaretos & Zemel, 2019), quantifies the minimum "cost" of transforming one probability distribution into another. For two probability distributions $P$ and $Q$ defined on a metric space $(M, d)$, the $p$-Wasserstein distance is formally defined as:

$$\mathcal{W}_p(P, Q) = \left( \inf_{\gamma \in \Gamma(P,Q)} \int_{M \times M} d(x,y)^p \, d\gamma(x,y) \right)^{1/p} \tag{42}$$

where $\Gamma(P, Q)$ denotes the set of all joint distributions $\gamma(x, y)$ whose marginals are $P$ and $Q$ respectively. In our evaluation, we focus on:

**1-Wasserstein Distance** ($\mathcal{W}_1$): This metric is particularly sensitive to differences in the locations of probability mass, making it suitable for detecting discrepancies in mode positioning.

**2-Wasserstein Distance** ($\mathcal{W}_2$): This metric accounts for both the locations and shapes of the distributions, providing a more comprehensive assessment of distributional similarity.

For practical computation with finite samples $\{x_i\}_{i=1}^n \sim P$ and $\{y_j\}_{j=1}^m \sim Q$, we employ the empirical approximation: $\mathcal{W}_p(\hat{P}_n, \hat{Q}_m) = \left( \min_{\pi \in \Pi(n,m)} \sum_{i=1}^n \sum_{j=1}^m \pi_{ij} \, d(x_i, y_j)^p \right)^{1/p}$, where $\Pi(n, m)$ is the set of all $n \times m$ transport plans $\pi$ with $\sum_j \pi_{ij} = 1/n$ and $\sum_i \pi_{ij} = 1/m$. We implement this computation using the POT (Python Optimal Transport) library (Flamary et al., 2021).

**Total Variation Distance.** The Total Variation Distance (TVD) between two probability distributions $P$ and $Q$ defined on the same sample space $\Omega$ is given by:

$$\text{TVD}(P, Q) = \sup_{A \subset \Omega} |P(A) - Q(A)| = \frac{1}{2} \int_\Omega |p(x) - q(x)| \, dx \tag{43}$$

where $p$ and $q$ are the probability density functions of $P$ and $Q$, respectively. For discrete distributions or histogrammed data, this simplifies to:

$$\text{TVD}(P, Q) = \frac{1}{2} \sum_i |P(A_i) - Q(A_i)| \tag{44}$$

where $\{A_i\}$ forms a partition of the sample space. In our evaluation framework, we apply the TVD to different aspects of the generated samples:

**Energy-based TVD (E-TVD)**: We construct normalized histograms of the log-energy values $\{\log E(x_i)\}$ for both the generated samples and reference samples, then compute the TVD between these histograms. This metric assesses how well our method captures the energy landscape of the target distribution.

**Sample-space TVD (S-TVD)**: For low-dimensional distributions like the GMM benchmark, we directly compute the TVD between normalized histograms of the generated and reference samples in the original sample space. We use adaptive binning to ensure accurate density estimation.

**Distance-based TVD (D-TVD)**: For high-dimensional particle systems, we transform the samples to a more interpretable representation by computing histograms of all pairwise interatomic distances

$\{d_{ij} = \|x_i - x_j\|_2\}$ for both generated and reference configurations. The TVD between these histograms evaluates the structural accuracy of the generated particle configurations.

For histogram-based TVD calculations, we employ a simple yet effective binning strategy where the number of bins is determined by the square root of the sample size: $n_{\text{bins}} = \lfloor \sqrt{n_{\text{samples}}} \rfloor$

### G.3 DETAILED INTRODUCTION ON BASELINES

**Path Integral Sampler(PIS) (Zhang & Chen, 2022).** PIS is a neural sampling approach that formulates sampling from unnormalized distributions as a stochastic optimal control problem. The method is built on the Schrödinger bridge framework, which aims to recover the most likely evolution of a diffusion process between initial and terminal distributions. PIS creates a controller parameterized by a neural network that guides particles from a simple prior distribution to the target distribution by minimizing the control energy while maximizing terminal likelihood. By leveraging the Girsanov theorem, PIS transforms the sampling problem into a control problem with a specific terminal cost. This approach enables free-form network architecture design without constraints on invertibility, unlike normalizing flows. The method can be trained end-to-end and provides theoretical guarantees on sample quality through the Wasserstein distance. A key limitation of PIS is that training requires simulating both forward and reverse trajectories, necessitating expensive backpropagation through the simulated paths, which can limit its scalability to high-dimensional problems compared to simulation-free approaches.

**Flow Annealed Importance Sampling Bootstrap (FAB) (Midgley et al., 2023b).** FAB is a novel approach for training normalizing flows to approximate complex multimodal target distributions without requiring samples from these distributions. FAB addresses the limitations of existing methods by combining $\alpha$-divergence minimization with an annealed importance sampling (AIS) (Neal, 2001) bootstrapping mechanism.

The objective function in FAB is explicitly formulated as minimizing the $\alpha$-divergence with $\alpha = 2$ between the target distribution $p$ and the flow model $q$:

$$D_{\alpha=2}(p||q) = -\frac{\int p(x)^\alpha q(x)^{1-\alpha} dx}{\alpha(1-\alpha)} = \int \frac{p(x)^2}{q(x)} dx$$

This objective favors mass-covering behavior while minimizing importance weight variance. To estimate this challenging objective, FAB uses AIS with the flow $q$ as the initial distribution and $p^2/q$ as the target distribution. By targeting this ratio, FAB focuses sampling on regions where the flow poorly approximates the target, providing higher-quality training signals.

The gradient of the loss function $L(\theta) \propto D_{\alpha=2}(p||q_\theta)$ with respect to the flow parameters $\theta$ is estimated as:

$$\nabla_\theta L(\theta) = -\mathbb{E}_{\text{AIS}}[w_{\text{AIS}}\nabla_\theta \log q_\theta(\bar{x}_{\text{AIS}})]$$

where $\bar{x}_{\text{AIS}}$ and $w_{\text{AIS}}$ are samples and importance weights generated by AIS. To reduce computational costs, FAB implements a prioritized replay buffer system that reuses AIS samples, significantly improving efficiency.

**iDEM (Akhound-Sadegh et al., 2024).** Iterated Denoising Energy Matching (iDEM) is a neural sampler for drawing samples from unnormalized Boltzmann distributions using solely the energy function and its gradient. iDEM employs a bi-level algorithmic structure: the inner loop uses a novel stochastic regression objective called Denoising Energy Matching (DEM) that directly targets the score function without requiring data samples, while the outer loop leverages diffusion to amortize sampling. By alternating between sampling regions of high model density and improving the sampler through stochastic matching, iDEM effectively explores complex energy landscapes. The approach is simulation-free in the inner loop, requiring no MCMC samples, and exploits the fast mode-mixing behavior of diffusion to smooth the energy landscape. While achieving excellent performance across various sampling tasks, iDEM's primary limitation lies in the bias of its estimator, corresponding to the no-correction version of the ideal objective in Eq. 6:

$$\mathbb{E}_{t\sim U(0,1)}\mathbb{E}_{x_t\sim p_t^{\mathcal{B}}}\left[\|s_\theta(x_t,t) - \nabla \log p_t(x_t)\|^2\right].$$

This objective can be viewed as a heuristic approach to achieve mode coverage since the sampled distribution $p_t^{\mathcal{B}}$ dynamically adjusts during training, whereas our importance correction version provides a more principled derivation from the forward KL between path measures.

**Diffusive KL Divergence (DiKL) (He et al., 2024).** DiKL offers a novel approach to training neural samplers that can efficiently generate samples from unnormalized target distributions in just one step. Unlike standard reverse KL divergence which suffers from mode-seeking behavior on multi-modal distributions, DiKL minimizes the reverse KL along diffusion trajectories of both model and target densities:

$$\text{DiKL}(p||q) \equiv \sum_{t=1}^{T} w(t)\text{KL}(p * k_t || q * k_t)$$

where $w(t)$ is a positive scalar weighting function and $\{k_1, \ldots, k_T\}$ is a set of (scaled) Gaussian convolution kernels. This approach allows the model to capture multiple modes in the target distribution by convex-smoothing the KL objective. The method uses a neural network as a deterministic mapping function from latent space to sample space, trained with a tractable gradient estimator that leverages denoising score matching and mixed score identity.

However, DiKL has notable limitations: Compared to the density-based neural solver, like FAB (Midgley et al., 2023b), the density $p_\theta(x)$ of the neural sampler is intractable, preventing importance re-weighting to correct potential biases; Compared to the score based neural solver, like iDEM (Akhound-Sadegh et al., 2024) and our method, the one-step generator has limited model flexibility, making it challenging to handle extremely complex distributions (such as LJ-55 systems); training stability issues can occur near the convergence; and posterior sampling during training can become a computational bottleneck for complicated target distributions where MCMC mixing is slow.

### G.4 Details on Experimental Setup

Here we elaborate on the comprehensive configuration used across our experimental evaluations. For our baseline methods FAB (Midgley et al., 2023b) [2], iDEM (Akhound-Sadegh et al., 2024) [3], and DiKL (He et al., 2024) [4], we strictly followed the parameter configurations from their official implementations. For PIS (Zhang & Chen, 2022), we adopted the implementation approach used in iDEM (Akhound-Sadegh et al., 2024). Following the experimental protocol established in iDEM (Akhound-Sadegh et al., 2024), we conducted three training runs with different random seeds for all methods, reporting results as mean±std in our tables. Our method's experimental configuration maintains consistency with iDEM's settings, including the VE-SDE type and noise schedule, without extensive hyperparameter tuning, only making appropriate adjustments to the SNIS sample quantity and replay buffer size. All neural network architectures were trained using the Adam optimizer on a single NVIDIA A40 GPU. The specific configurations of our method for individual experiments are detailed below:

**GMM Benchmarks.** All architectures utilize MLPs incorporating sinusoidal and positional encodings. The network architecture consists of 3 hidden layers with dimensionality 128, complemented by positional embeddings also of dimension 128. During the training, generated samples were constrained to the interval [-1, 1] and subsequently rescaled by factors of 50, 100, and 150 for energy computation in GMM40, GMM80, and GMM120 respectively. VE-SDE proceeded with a geometric noise progression where $\sigma_{\min} = 1e - 5, \sigma_{\max} = 1$, utilizing $L = 500$ sample points for the regression target $\mathcal{S}_L$, with gradient norm constrained to a maximum value of 70. Neural networks were optimized using a learning rate of $5e - 4$. To balance training efficiency and performance, we set the SNIS quantity to 5 across all GMM benchmarks, with buffer sizes of 10k, 20k, and 10k for GMM40, GMM80, and GMM120 respectively.

---

[2] For GMM tasks, we use: https://github.com/lollcat/fab-torch. For particle system tasks, we use: https://github.com/lollcat/se3-augmented-coupling-flows

[3] https://github.com/jarridrb/DEM

[4] https://github.com/jiajunhe98/DiKL

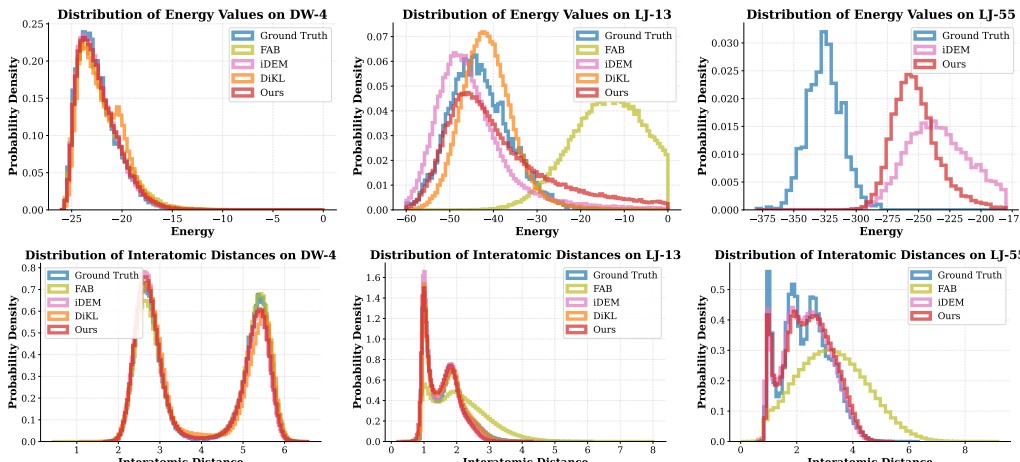

Figure 5: Energy and Interatomic Distance distributions for different molecular systems.

**Particle Systems.** For DW-4, we incorporated EGNNs with 3 message-passing iterations and a 2-layer MLP with width 128. VE-SDE was trained using a geometric noise progression with $\sigma_{min} = 1e-5$ and $\sigma_{max} = 3$, optimized at a learning rate of $1e-3$, with $L = 1000$ points for computing the regression target $\mathcal{S}_L$ and gradient norm capped at 20. The SNIS sample quantity was set to 2 for DW4 benchmark. For LJ-13, we employed EGNNs with 5 hidden layers at width 128. VE-SDE was trained with a geometric noise progression defined by $\sigma_{min} = 0.01$ and $\sigma_{max} = 2$, optimized at a learning rate of $1e-3$, utilizing $L = 1000$ points for the regression target $\mathcal{S}_L$ with gradient norm limited to 20. For LJ-55, we utilized EGNNs with 5 hidden layers at width 128. VE-SDE was trained with a geometric noise progression characterized by $\sigma_{min} = 0.5$ and $\sigma_{max} = 4$, optimized at a learning rate of $1e-3$, with $L = 100$ points for the regression target $\mathcal{S}_L$ and gradient norm constrained to 20.

### G.5 ADDITIONAL VISUALIZATION RESULTS

Figure 5 presents the performance of our method on particle system benchmarks of increasing complexity: the 4-particle Double-Well potential (DW-4), the 13-particle Lennard-Jones cluster (LJ-13), and the challenging 55-particle Lennard-Jones cluster (LJ-55). The E-TVD metric and corresponding visualizations in Figure 5 (top row) reveal how accurately each method captures the underlying energy landscape. For DW-4, all methods produce reasonable energy distributions. The LJ-13 system presents a more significant challenge, particularly for FAB, which produces a dramatically shifted distribution centered around -20 energy units rather than the reference distribution centered around -45 units. Both DiKL and iDEM capture the energy distribution more accurately but with noticeable deviations in peak height and tail behavior. Our method achieves the closest match to the ground truth energy distribution shape. For the LJ-55 benchmark, the visualization reveals a significant energy distribution shift for both iDEM and our method compared to ground truth, though our approach produces a distribution with a shape more closely resembling the reference. The interatomic distance distributions (Figure 5, bottom row) provides insight into how well each method captures the structural properties of the particle systems. For DW-4, all methods successfully capture the bimodal structure visible in the distance distribution. In the LJ-13 system, our method better captures the critical first and second peak heights that correspond to nearest and second-nearest neighbor distances in the molecular structure compared to FAB. For the challenging LJ-55 system, our method successfully reproduces the multi-peak structure of the reference distribution with higher fidelity, while FAB exhibits substantial deviations in peak locations and relative heights.

### G.6 NESS ANALYSIS IN HIGH-DIMENSIONAL SETTINGS

Figure 6 shows the nESS behavior for high-dimensional molecular systems DW4 and LJ-13. Notably, the nESS values are substantially lower than those observed for GMM benchmarks: DW4 stabilizes around 0.70 for SNIS-2 and 0.50 for SNIS-5, while LJ-13 reaches approximately 0.58 for SNIS-2 and

0.28 for SNIS-5. We observe a clear trend of decreasing nESS values as the problem dimensionality increases—from GMM benchmarks (2D) to DW4 (8D) to LJ-13 (39D).

These lower nESS values in high dimensions are expected and, importantly, do not indicate method failure. Instead, they reflect a fundamental characteristic of importance sampling in challenging settings: the method naturally identifies and emphasizes the most informative samples from the proposal distribution. In high-dimensional spaces where the proposal $p_t^{\mathcal{B}}$ may significantly differ from the target $p_t$ in certain regions, the importance weights become more selective: a few samples that better represent the target distribution receive higher weights, while less representative samples receive lower weights. This selectivity is actually beneficial for our training objective. By assigning higher importance to samples that are more representative of the true distribution, our method effectively focuses the learning signal on the most informative data points, concentrating on those that provide the strongest gradient signal for improving the score network.

**DW4 and LJ-13: Average nESS During Training**

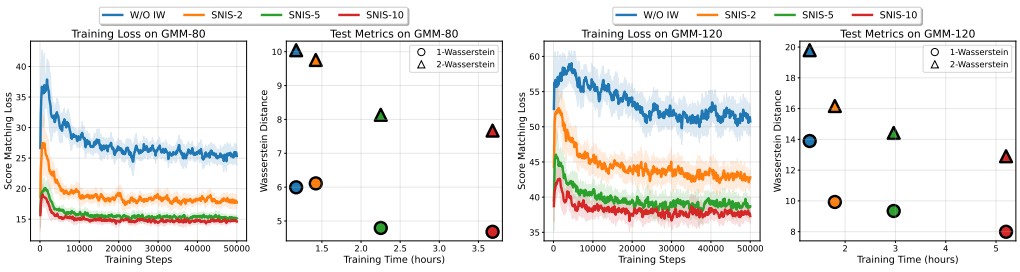

Figure 6: nESS evolution for high-dimensional molecular systems DW4 and LJ-13. Lower nESS values compared to GMM benchmarks reflect the increased selectivity of importance weights in challenging high-dimensional spaces.

Furthermore, the nESS also serves as an indirect measure of the similarity between the target distribution $p_t$ and the proposal distribution $p_t^{\mathcal{B}}$. As evidenced in Tables 1 and 2, the distribution discrepancy metrics increase with problem complexity and dimensionality. This growing discrepancy between the learned and target distributions naturally manifests as lower nESS values, reflecting the increased challenge of approximating high-dimensional, multi-modal distributions.

Despite these lower nESS values in challenging settings, our method consistently achieves the best performance compared to all baseline approaches across both GMM and molecular benchmarks. This demonstrates a crucial insight: effective importance weighting does not require uniformly balanced weights. Rather than serving merely as a variance reduction technique, SNIS acts as an intelligent sample selection mechanism that identifies which historical samples are most valuable for current training. Even when this selection results in heavily weighting a subset of proposals (low nESS), it successfully focuses the learning signal on the most informative samples, ultimately leading to superior sample quality. The effectiveness of this selective weighting strategy is validated by our method's consistent state-of-the-art performance across all evaluated metrics.

### G.7 Additional Results of Importance Weight Correction

Figure 7 presents training and test results for our method's importance sampling variants on the more complex GMM-80 and GMM-120 benchmarks.

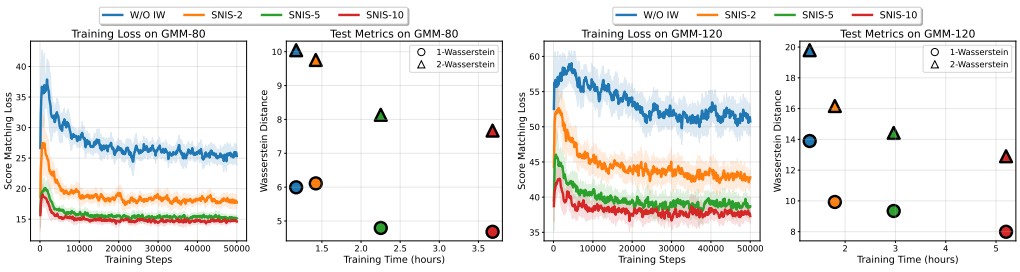

Figure 7: Comparison of training and testing performance on the GMM-80 and GMM-120. Left: Training loss trajectories for different importance sampling strategies. Right: Evaluation of trained models using 1-Wasserstein (circles) and 2-Wasserstein (triangles) distance metrics.

Similar to the GMM-40 results, the training loss plots (Figure 7, Left) for both GMM-80 and GMM-120 show that all SNIS variants achieve significantly lower final training losses compared to the W/O IW baseline. The SNIS loss trajectories are also smoother and converge faster, particularly evident on the more complex GMM-120. Within the SNIS variants, using more proposal samples (SNIS-5, SNIS-10) generally leads to slightly lower losses and improved stability compared to SNIS-2.

The test metric plots (Figure 7, Right) confirm that these training improvements translate to better sample quality. On GMM-80, SNIS-2 significantly reduces Wasserstein distances ($\mathcal{W}_1$ and $\mathcal{W}_2$) compared to the baseline, with further substantial improvements seen with SNIS-5. SNIS-10 achieves metrics very close to SNIS-5, suggesting diminishing returns on performance from additional samples beyond a certain point, similar to GMM-40. The training times increase with the number of proposal samples, illustrating the performance-cost trade-off. On GMM-120, the performance differences are even more pronounced; while the baseline and SNIS-2 struggle to achieve low Wasserstein distances, SNIS-5 and SNIS-10 demonstrate significantly better sample quality, highlighting the necessity of robust importance weighting for complex multimodal targets. SNIS-10 achieves slightly better metrics than SNIS-5 on GMM-120 at the cost of longer training time.

## H JUSTIFICATION FOR ENHANCED MODE COVERAGE BY FORWARD KL DIVERGENCE

In this section, we argue why optimizing our weighted objective is theoretically better suited for achieving comprehensive mode coverage. The key difference arises from how the optimization process treats regions where the true distribution $p_t$ and the neural sampler-induced proposal distribution $p_t^{\mathcal{B}}$ significantly diverge.

### H.1 GRADIENT INTERPRETATION: THE ROLE OF IMPORTANCE WEIGHTS

Let us examine the expected gradient updates provided by the weighted versus unweighted objectives. Ignoring the complexities introduced by the biases of $S_L$ and the SNIS estimation for this qualitative argument, the gradient corresponding to our ideal weighted objective behaves approximately as:

$$\nabla_\theta L_{\text{ideal}} \approx \nabla_\theta \mathbb{E}_{t,p_t^{\mathcal{B}}}[w(x_t,t)\|s_\theta(x_t,t) - \nabla \log p_t(x_t)\|^2] \tag{45}$$

$$\approx \mathbb{E}_{t,p_t^{\mathcal{B}}}[w(x_t,t) \cdot 2(s_\theta(x_t,t) - \nabla \log p_t(x_t)) \cdot \nabla_\theta s_\theta(x_t,t)], \tag{46}$$

where $w(x_t,t) = p_t(x_t)/p_t^{\mathcal{B}}(x_t)$ is the true importance weight. In our practical algorithm, $w$ is approximated by $w_{\text{norm}} \propto \tilde{w} \propto Zw$. Contrast this with the gradient of the unweighted objective:

$$\nabla_\theta L_{\text{unweighted}} \approx \nabla_\theta \mathbb{E}_{t,p_t^{\mathcal{B}}}[\|s_\theta(x_t,t) - \nabla \log p_t(x_t)\|^2] \tag{47}$$

$$\approx \mathbb{E}_{t,p_t^{\mathcal{B}}}[2(s_\theta(x_t,t) - \nabla \log p_t(x_t)) \cdot \nabla_\theta s_\theta(x_t,t)]. \tag{48}$$

The crucial difference lies in the presence of the importance weight $w(x_t,t)$ (approximated by $w_{\text{norm}}$) inside the expectation of Eq. 45. Consider a region in the state space corresponding to a mode of the true target $\pi$ that is currently under-represented in the buffer $\mathcal{B}$. For samples $x_t$ originating from this region, $p_t(x_t)$ will be relatively high, while the proposal density $p_t^{\mathcal{B}}(x_t)$ will be relatively low. Consequently, the true importance weight $w(x_t,t)$ will be large in this region. Our estimator $\tilde{w}$, being proportional to $Zw$, will also tend to yield larger values here compared to regions well-represented in the buffer. This provides a strong learning signal, pushing the parameters $\theta$ to adjust $s_\theta$ specifically to reduce errors in these critical regions identified by high importance weights.

Conversely, the unweighted gradient (Eq. 47) treats the error contribution from all samples drawn from $p_t^{\mathcal{B}}$ equally. Errors occurring in low-density regions of $p_t^{\mathcal{B}}$, even if they correspond to high-density modes of $p_t$, will only contribute significantly to the gradient if they are sampled frequently, which is unlikely if the buffer poorly represents those modes. The unweighted objective therefore lacks a direct mechanism to prioritize learning in under-represented areas identified by the true distribution. Thus, the importance weighting actively redirects the optimization towards matching the score function according to the structure of $p_t$, not just $p_t^{\mathcal{B}}$.

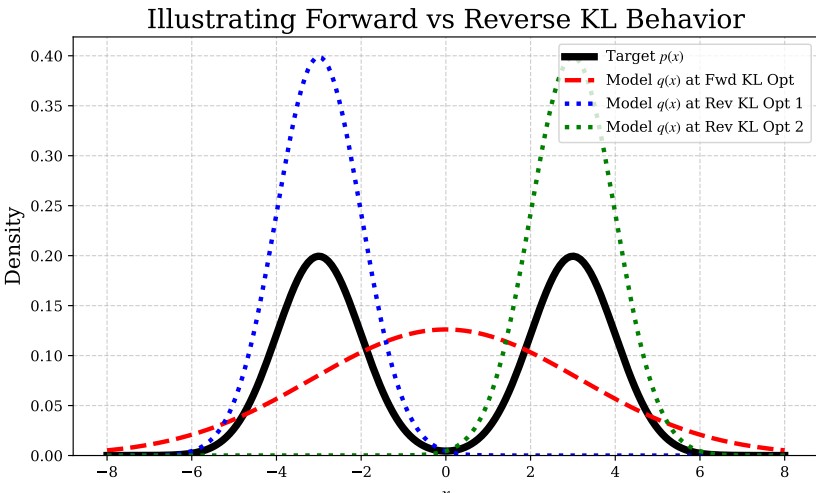

Figure 8: Illustration of Forward vs. Reverse KL divergence behavior on a 1D bimodal Gaussian mixture target distribution. The black curve shows the target distribution. The red dashed curve shows the unimodal Gaussian model positioned at the mean that minimizes the Forward KL divergence (mode-averaging). The blue and green dotted curves show the unimodal Gaussian model positioned at the means that approximate the two local minima of the Reverse KL divergence (mode-seeking).

## H.2 AN ILLUSTRATIVE EXAMPLE

The fundamental difference between forward and reverse KL objectives in terms of mode handling can be clearly illustrated with a simple toy example. Consider a one-dimensional target distribution $p(x)$ composed of an equal mixture of two Gaussians with shared variance $\sigma^2$ but distinct means $\mu_1$ and $\mu_2$: $p(x) = 0.5\mathcal{N}(x; \mu_1, \sigma^2) + 0.5\mathcal{N}(x; \mu_2, \sigma^2)$. Let us attempt to approximate this bimodal $p(x)$ using a unimodal Gaussian model $q(x; \mu, \hat{\sigma}) = \mathcal{N}(x; \mu, \hat{\sigma}^2)$, optimizing the mean $\mu$ and scale $\hat{\sigma}$.

**Forward KL Optimization ($\mathrm{KL}(p \,\|\, q)$).** Minimizing forward KL divergence $\mathrm{KL}(p \,\|\, q) = \int p(x) \log \frac{p(x)}{q(x; \mu, \hat{\sigma}^2)} dx$ with respect to $\mu$ and $\hat{\sigma}^2$ is equivalent to maximizing the expected log-likelihood of the model under the target distribution:

$$(\mu^*, \hat{\sigma}^{2*}) = \arg\max_{\mu, \hat{\sigma}^2} \int p(x) \log q(x; \mu, \hat{\sigma}^2) dx = \arg\max_{\mu, \hat{\sigma}^2} \mathbb{E}_{p(x)}[\log q(x; \mu, \hat{\sigma}^2)]$$

where the log-likelihood of the Gaussian model is: $\log q(x; \mu, \hat{\sigma}^2) = -\frac{(x-\mu)^2}{2\hat{\sigma}^2} - \frac{1}{2}\log(2\pi\hat{\sigma}^2)$. We maximize this objective by setting the partial derivatives with respect to $\mu$ and $\hat{\sigma}^2$ to zero.

First, optimize with respect to $\mu$ for a fixed $\hat{\sigma}^2$. Maximizing $\mathbb{E}_{p(x)}\left[-\frac{(x-\mu)^2}{2\hat{\sigma}^2}\right]$ is equivalent to minimizing $\mathbb{E}_{p(x)}[(x-\mu)^2]$, as other terms are constant with respect to $\mu$ or $\hat{\sigma}^2$. The minimum of $\mathbb{E}_{p(x)}[(x-\mu)^2]$ occurs at $\mu = \mathbb{E}_{p(x)}[x]$. For $p(x) = 0.5\mathcal{N}(x; \mu_1, \sigma^2) + 0.5\mathcal{N}(x; \mu_2, \sigma^2)$:

$$\mu^* = \mathbb{E}_{p(x)}[x] = 0.5\mathbb{E}_{\mathcal{N}(x;\mu_1,\sigma^2)}[x] + 0.5\mathbb{E}_{\mathcal{N}(x;\mu_2,\sigma^2)}[x] = \frac{\mu_1 + \mu_2}{2}.$$

Next, optimize with respect to $\hat{\sigma}^2$ for a fixed $\mu$. Setting the derivative with respect to $\hat{\sigma}^2$ to zero:

$$\frac{\partial}{\partial \hat{\sigma}^2} \mathbb{E}_{p(x)}\left[-\frac{(x-\mu)^2}{2\hat{\sigma}^2} - \frac{1}{2}\log(2\pi\hat{\sigma}^2)\right] = \frac{1}{2(\hat{\sigma}^2)^2}\mathbb{E}_{p(x)}[(x-\mu)^2] - \frac{1}{2\hat{\sigma}^2} = 0.$$

Assuming $\hat{\sigma}^2 > 0$, we multiply by $2(\hat{\sigma}^2)^2$:

$$\mathbb{E}_{p(x)}[(x-\mu)^2] - \hat{\sigma}^2 = 0 \implies \hat{\sigma}^{2*} = \mathbb{E}_{p(x)}[(x-\mu)^2].$$

Substituting the optimal $\mu^* = \mathbb{E}_{p(x)}[x]$ into the equation for $\hat{\sigma}^{2*}$ yields the variance of $x$ under $p(x)$:

$$\hat{\sigma}^{2*} = \mathbb{E}_{p(x)}[(x - \mathbb{E}_{p(x)}[x])^2] = \text{Var}_{p(x)}(x).$$

The variance of $p(x)$ can be calculated as $\text{Var}_{p(x)}(x) = \mathbb{E}_{p(x)}[x^2] - (\mathbb{E}_{p(x)}[x])^2$. We have $\mathbb{E}_{p(x)}[x] = \frac{\mu_1 + \mu_2}{2}$ and $\mathbb{E}_{p(x)}[x^2] = \sigma^2 + 0.5(\mu_1^2 + \mu_2^2)$ (this follows from $\mathbb{E}[X^2] = \text{Var}(X) + (\mathbb{E}[X])^2$ for each component).

$$\hat{\sigma}^{2*} = \left(\sigma^2 + 0.5(\mu_1^2 + \mu_2^2)\right) - \left(\frac{\mu_1 + \mu_2}{2}\right)^2 = \sigma^2 + \frac{(\mu_1 - \mu_2)^2}{4}.$$

**Reverse KL Optimization ($\text{KL}(q \,\|\, p)$).**  Minimizing $\text{KL}(q \,\|\, p) = \int q(x; \mu, \hat{\sigma}^2) \log \frac{q(x; \mu, \hat{\sigma}^2)}{p(x)} dx$ with respect to $\mu$ and $\hat{\sigma}^2$. The optimal conditions are $\nabla_\mu \text{KL}(q\|p) = 0$ and $\nabla_{\hat{\sigma}^2} \text{KL}(q\|p) = 0$.

Using the derivatives derived previously, the conditions are:

$$\mathbb{E}_{q(x;\mu,\hat{\sigma}^2)}\left[\frac{x - \mu}{\hat{\sigma}^2}(\log q(x; \mu, \hat{\sigma}^2) - \log p(x))\right] = 0,$$

$$\mathbb{E}_{q(x;\mu,\hat{\sigma}^2)}\left[\left(\frac{(x - \mu)^2}{2(\hat{\sigma}^2)^2} - \frac{1}{2\hat{\sigma}^2}\right)(\log q(x; \mu, \hat{\sigma}^2) - \log p(x))\right] = 0.$$

These simplify to:

$$\mathbb{E}_{q(x;\mu,\hat{\sigma}^2)}[(x - \mu) \log p(x)] = \mathbb{E}_{q(x;\mu,\hat{\sigma}^2)}[(x - \mu) \log q(x; \mu, \hat{\sigma}^2)] = 0,$$

$$\mathbb{E}_{q(x;\mu,\hat{\sigma}^2)}\left[\left(\frac{(x - \mu)^2}{\hat{\sigma}^2} - 1\right) \log p(x)\right] = \mathbb{E}_{q(x;\mu,\hat{\sigma}^2)}\left[\left(\frac{(x - \mu)^2}{\hat{\sigma}^2} - 1\right) \log q(x; \mu, \hat{\sigma}^2)\right] = 0.$$

Substituting $q(x; \mu, \hat{\sigma}^2) = \mathcal{N}(x; \mu, \hat{\sigma}^2)$ and $p(x) = 0.5\mathcal{N}(x; \mu_1, \sigma^2) + 0.5\mathcal{N}(x; \mu_2, \sigma^2)$ yields a coupled system of transcendental equations for $\mu$ and $\hat{\sigma}^2$.

Solving this system analytically is generally intractable. However, qualitative analysis and numerical studies show that for sufficiently separated modes of $p(x)$ (relative to $\sigma$), the optimization landscape of $\text{KL}(q \,\|\, p)$ will still exhibit local minima where the unimodal Gaussian $q$ collapses onto one of the modes of $p$. The local minima will be approximately at parameters:

$$(\mu, \hat{\sigma}^2) \approx (\mu_1, \sigma^2) \quad \text{and} \quad (\mu, \hat{\sigma}^2) \approx (\mu_2, \sigma^2).$$

At these minima, the unimodal Gaussian $q$ aligns its mean and variance with one of the components of the bimodal target $p$, achieving a locally low KL divergence value by fitting one part of the target distribution well, while effectively ignoring the other part. The point $(\mu = (\mu_1 + \mu_2)/2, \hat{\sigma}^2 = \text{Var}_{p(x)}(x))$ which is optimal for Forward KL is typically a local maximum or a saddle point for Reverse KL. This behavior is a well-known property of Reverse KL when the model is less expressive than the target distribution (Bishop & Nasrabadi, 2006).

**Difference in Optimization Results.**  The core difference in the optimal solutions found by minimizing Forward KL versus Reverse KL is stark in this scenario:

- **Forward KL** yields a single optimal unimodal Gaussian that *covers* both modes by centering at the overall mean and having the total variance of the target distribution.

- **Reverse KL** yields local optima where the unimodal Gaussian *collapses* onto one specific mode of the target distribution. The optimal Gaussian is centered near one of the target modes and has a variance similar to the variance of that mode ($\sigma^2$). It misses the other modes entirely.

This example with learnable variance further highlights that Forward KL tends towards mode coverage (finds a single model that spans the support of the data), while Reverse KL tends towards mode-seeking (finds a model that fits one high-density region of the data well, potentially missing others), making Forward KL-related objectives more suitable for tasks like generative modeling where capturing all modes of the data distribution is desired.

Table 3: Comparison to direct sampling with estimated scores across Gaussian mixture models of varying complexity. Metrics include 1-Wasserstein ($\mathcal{W}_1$) and 2-Wasserstein ($\mathcal{W}_2$) distances, energy total variation distance (E-TVD), and sample level total variation distance (S-TVD). Values reported as mean±std across 3 random seeds.

| Method | GMM-40 | | | | GMM-80 | | | | GMM-120 | | | |
|---|---|---|---|---|---|---|---|---|---|---|---|---|
| | $\mathcal{W}_1$ | $\mathcal{W}_2$ | E-TVD | S-TVD | $\mathcal{W}_1$ | $\mathcal{W}_2$ | E-TVD | S-TVD | $\mathcal{W}_1$ | $\mathcal{W}_2$ | E-TVD | S-TVD |
| Ours | 1.43±0.6 | 3.12±1.3 | 0.05±0.0 | **0.19**±0.0 | 3.21±0.4 | **6.58**±0.7 | 0.12±0.0 | **0.21**±0.0 | **5.05**±0.9 | **9.90**±1.2 | 0.23±0.0 | 0.30±0.0 |
| DwES (L=500) | 1.61±0.3 | 3.39±0.4 | 0.05±0.0 | 0.29±0.0 | 6.02±0.3 | 10.46±0.4 | 0.08±0.0 | 0.38±0.0 | 64.04±9.9 | 73.88±11.6 | 0.76±0.1 | 0.99±0.0 |
| DwES (L=1000) | **1.23**±0.2 | **2.81**±0.3 | 0.05±0.0 | 0.28±0.0 | 4.05±0.4 | 7.94±0.8 | 0.06±0.0 | 0.28±0.0 | 11.22±1.8 | 18.03±2.5 | 0.12±0.0 | 0.54±0.0 |
| DwES (L=2000) | 1.24±0.1 | 2.97±0.3 | **0.04**±0.0 | 0.27±0.0 | **3.20**±0.3 | 6.97±0.4 | **0.05**±0.0 | 0.25±0.0 | 7.37±0.3 | 13.02±0.7 | **0.06**±0.0 | **0.29**±0.0 |

# I  DIRECT SAMPLING WITH ESTIMATED SCORES

Given the target distribution's score estimator developed in section D, we can explore an alternative sampling approach that directly leverages this estimator within the reverse SDE framework (Huang et al., 2023; Grenioux et al., 2024) .

The true reverse SDE that generates samples from our target distribution is given by:

$$dx_t = \left[ f(x_t, t) - g(t)^2 \nabla \log p_t(x_t) \right] dt + g(t)d\bar{w}_t,$$

where $\nabla \log p_t(x_t)$ represents the score of the marginal distribution at time $t$. Using VE-SDE and the Monte Carlo estimator in Eq. 7, we can approximate this score as:

$$\nabla \log p_t(x_t) \approx S_L(x_t, t) := \nabla \log \sum_{i=1}^{L} \exp(-E(x_{0|t}^{(i)})), \quad \text{where } \{x_{0|t}^{(i)}\}_{i=1}^{L} \sim \mathcal{N}(x_0; x_t, \sigma_t^2 I).$$

This allows us to formulate an approximate reverse SDE for sampling:

$$d\tilde{x}_t = \left[ f(\tilde{x}_t, t) - g(t)^2 S_L(\tilde{x}_t) \right] dt + g(t)d\bar{w}_t.$$

By numerically integrating this equation from $\tilde{x}_1 \sim \mathcal{N}(0, I)$ backward in time until $t = 0$, we obtain samples $\tilde{x}_0$ that approximate our target distribution. We conducted experiments to evaluate the quality of samples generated through this direct approach compared to our main method. While the direct estimated-score approach theoretically converges to the true distribution as $L \to \infty$, practical considerations arise with finite computational resources.

To evaluate the direct sampling approach (which we designate as Directly Sampling with Estimated Scores, or **DwES**), we maintain identical SDE configurations as used in our neural sampler experiments (detailed in Appendix G.4). For our ablation study on the impact of sample quantity in Monte Carlo estimation, we vary the parameter $L \in \{500, 1000, 2000\}$, generating 10,000 samples for each configuration on GMM-40, GMM-80 and GMM-120 benchmarks.

We assess sample quality using the same evaluation framework established in the main text, employing Wasserstein distances ($\mathcal{W}_1$ and $\mathcal{W}_2$) and Total Variation distances (further elaborated in Appendix G.2). All sampling experiments were repeated three times using different random seeds, with results reported as mean($\pm$std) to ensure robust evaluation. Additionally, we conduct a comparative runtime analysis, measuring the total wall-clock time required for DwES versus our proposed neural sampler approach under identical computational conditions. Tables 3 and 4 present a comprehensive comparison between our proposed method and DwES across GMM benchmarks of varying complexity. Figure 9 provides visual confirmation of these quantitative results.

**Sample Quality.** For the simplest GMM-40 benchmark, DwES with sufficient Monte Carlo samples ($L \geq 1000$) achieves comparable or slightly better sample quality than our method across most metrics. Specifically, DwES ($L = 1000$) achieves the best Wasserstein distances, while DwES ($L = 2000$) yields the lowest E-TVD score. However, our method maintains a superior S-TVD score (0.19 versus 0.27-0.29), indicating better overall probability density matching despite minor differences in other metrics. As distribution complexity increases, the advantage of our method becomes more pronounced. For GMM-80, only DwES with $L = 2000$ matches our method's performance on Wasserstein metrics, while achieving better E-TVD (0.05 versus 0.12) but worse S-TVD (0.25 versus 0.21). The visual comparison in Figure 9 (middle row) reveals that our method

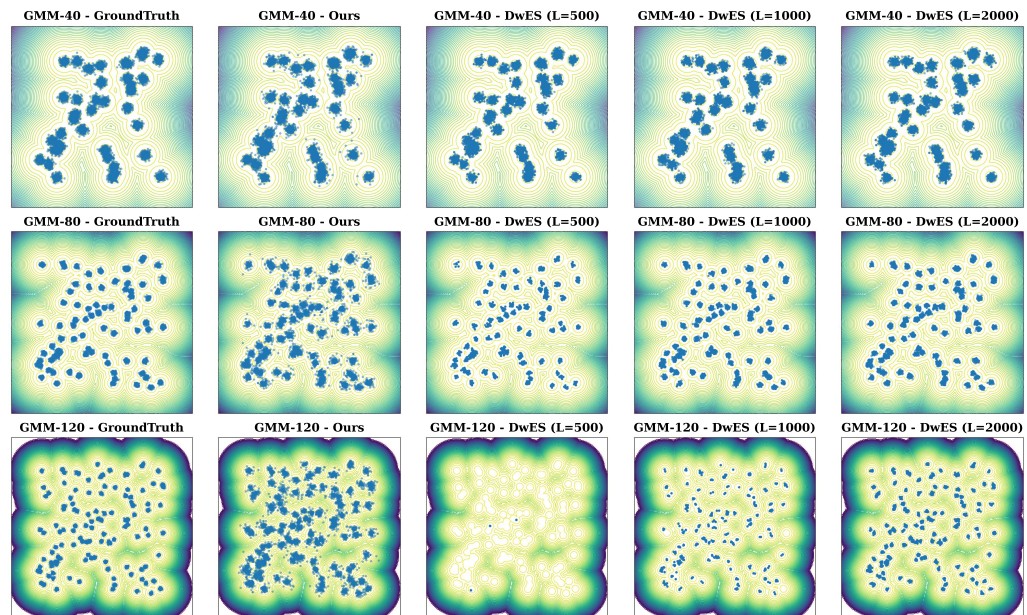

Figure 9: Visualization of samples from direct sampling with estimated scores across GMM benchmarks with increasing modes.

produces more uniformly distributed samples across all modes compared to DwES variants. The difference becomes dramatically apparent for the most challenging GMM-120 benchmark. DwES with limited Monte Carlo samples ($L = 500$) completely fails to capture the distribution structure, as evidenced by the extremely poor metric scores and the nearly uniform sample distribution shown in Figure 9 (bottom-middle). Even with $L = 2000$, DwES achieves substantially worse Wasserstein distances than our method, despite better E-TVD scores.

**Computational Efficiency.** Table 4 reveals the substantial computational advantage of our neural sampler approach. Even the most efficient DwES configuration ($L = 500$) requires approximately 72× more computation time than our method for GMM-40, scaling to 198× for GMM-120. The most accurate DwES variant ($L = 2000$) demands a staggering 286× more computation time for GMM-40 and 788× more for GMM-120. This computational disparity becomes increasingly prohibitive as distribution complexity grows, highlighting the efficiency benefits of amortizing the score estimation through neural network training.

Table 4: Comparison of computational time (in seconds) required to generate 10,000 samples across Gaussian mixture models. Values reported as mean±std across 3 random seeds.

| Method | GMM-40 | GMM-80 | GMM-120 |
|---|---|---|---|
| Ours | $1.45_{\pm 0.03}$ | $1.46_{\pm 0.05}$ | $1.55_{\pm 0.23}$ |
| DwES (L=500) | $104.78_{\pm 0.39}$ | $207.76_{\pm 0.25}$ | $307.55_{\pm 0.91}$ |
| DwES (L=1000) | $207.70_{\pm 0.44}$ | $410.95_{\pm 0.96}$ | $650.62_{\pm 10.35}$ |
| DwES (L=2000) | $414.40_{\pm 0.25}$ | $834.51_{\pm 10.44}$ | $1222.06_{\pm 3.07}$ |

**Complexity Scaling.** A critical observation is how the performance gap widens with increasing distribution complexity. While DwES can achieve competitive results for simpler distributions given sufficient Monte Carlo samples, its performance deteriorates rapidly for higher-mode-count distributions. Our method maintains consistent performance across all benchmarks, demonstrating superior scalability to complex target distributions.

**Practical Implications.** These results demonstrate the fundamental trade-off between direct estimated-score sampling and neural sampling approaches. DwES can potentially achieve arbitrary accuracy given sufficient computational resources (large enough $L$), but the computational cost becomes prohibitive for complex distributions. The neural sampling approach effectively amortizes

this cost by learning the score function during training, enabling significantly faster sampling at inference time while maintaining competitive or superior sample quality.

## J    LIMITATIONS, FUTURE WORKS AND BROADER IMPACT

This section discusses the current limitations of the proposed *Importance Weighted Score Matching* method and reflects on its potential broader societal impacts.

### J.1    LIMITATIONS AND FUTURE WORK

Despite the promising results presented in this work, our method, like many advanced generative modeling techniques, faces certain challenges and limitations that offer avenues for future research:

**Scalability to Very High Dimensions:** While our method demonstrates strong performance on complex, multi-modal problems, effectively scaling neural samplers to extremely high-dimensional systems, such as the Lennard-Jones 55-particle cluster (LJ55) mentioned in our experiments, remains a significant hurdle. The computational cost of both training the score model and generating samples via the reverse SDE can become prohibitive. Future work could explore more efficient network architectures, SDE solvers tailored for high dimensions, or hierarchical modeling approaches.

**Bias-Variance Trade-off in Importance Sampling:** The use of self-normalized importance sampling (SNIS) is central to our method. However, SNIS estimators are known to possess a bias-variance trade-off that can be intricate. While our theoretical analysis provides bounds, a deeper investigation into this trade-off, particularly concerning the choice and quality of the proposal distribution $p_t^{\mathcal{B}}(x_t)$ and the number of importance samples, could lead to strategies for further reducing variance, mitigating bias, and improving overall training efficiency. This might also yield refined theoretical guarantees under more specific conditions.

**Sampling Speed of Diffusion Models:** Generating samples from trained diffusion models typically involves simulating a reverse-time SDE or ODE, which requires multiple (often hundreds or thousands) of function evaluations of the learned score model $s_\theta$. This iterative process can be slow, hindering practical deployment in applications demanding rapid sample generation. Accelerating the sampling process, for instance, through techniques like progressive distillation (Salimans & Ho, 2022), consistency models (Song et al., 2023), or other fast inference strategies specifically adapted for our method, is an essential direction for future work.

**Choice of Proposal Distribution for IS:** The effectiveness of our method relies on the quality of the proposal distribution $p_t^{\mathcal{B}}(x_t)$ used for importance sampling. While we employed standard choices, adaptive or learned proposal distributions could potentially improve performance, reduce variance, and enhance robustness, especially for targets with highly complex or disparate modes. Developing such strategies within the proposed framework is a promising research avenue.

Addressing these limitations will be crucial for advancing the applicability and robustness of our method and similar methods for sampling from unnormalized densities.

### J.2    BROADER IMPACT

The ability to efficiently and accurately sample from unnormalized probability distributions without direct target samples has significant implications across various scientific and engineering disciplines.

**Potential Positive Impacts:** *Scientific Discovery:* Many problems in physics (e.g., statistical mechanics, condensed matter), chemistry (e.g., molecular dynamics, drug discovery), and biology (e.g., protein folding, systems biology) involve characterizing complex systems described by unnormalized energy functions or likelihoods. our method could provide a powerful tool for exploring configuration spaces, estimating thermodynamic properties, and generating hypotheses in these domains, potentially accelerating discovery. *Machine Learning and AI:* Beyond direct scientific applications, generating diverse and high-quality samples is crucial for areas like reinforcement learning (e.g., policy exploration), generative art, data augmentation, and robust AI systems. The mode-coverage capabilities of our method could be particularly beneficial. *Optimization:* Sampling methods can be used to explore complex, non-convex landscapes in optimization problems. By effectively covering

multiple modes, our method might inspire new approaches to global optimization or finding diverse sets of solutions.

**Potential Negative Impacts and Ethical Considerations:** Training large diffusion models and performing extensive importance sampling can be computationally intensive, contributing to energy consumption and carbon footprint. Efforts towards algorithmic efficiency, as discussed in "Future Work," are important not only for practicality but also for sustainability.

## LLM USAGE STATEMENT

In preparing this manuscript, we utilized large language models as a writing assistance tool for polishing, rearranging, and proofreading the text. The LLM was used to improve clarity, correct grammatical errors, and enhance the overall readability of the manuscript. All scientific ideas, research methodology, experimental design, results analysis, and core intellectual contributions were developed by the authors without LLM assistance.

