# OpenReview forum: "Importance Weighted Score Matching for Diffusion Samplers with Enhanced Mode Coverage"
_ICLR.cc/2026/Conference — ICLR 2026 Conference Withdrawn Submission_

### Official Review · Reviewer_i4Gt · 2025-10-17

**Soundness:** 1
**Presentation:** 2
**Contribution:** 1
**Rating:** 2
**Confidence:** 4

**Summary:**

this paper aims to address the problem of sampling from unnormalised density function.
It proposed a modification to idem by adding an importance sampling for Target score matching, which showing better mode coverage and overall performance in GMM, DW and LJ systems.

**Strengths:**

1. The writing is generally easy to follow.
2. The performance of the algorithm, compared to FAB, IDEM. DIKL, is promising.
3. The paper not only provide empirical results, but also derive theorical bounds for the proposed algorithm.

**Weaknesses:**

1. Limited contribution; key limitations unaddressed. The core contribution of this paper is to estimate an importance weight for the score matching objective, which is an natural but limited extension to iDEM.
iDEM suffers from (i) high-variance score estimates under the target-score identity, (ii) inefficiency in terms of energy evaluations, and (iii) mode-balance issues (i.e., for multi-mode target, the obtained sample cannot reflect the true weight for these modes). The proposed method does not convincingly resolve any of these. Moreover, while the paper claims improved mode coverage, this is not a fundamental weakness of iDEM on sufficiently smooth targets. Because iDEM employs importance sampling with a Gaussian proposal proportional to the noising kernel, it can already discover nearby missing modes; also, importance sampling alone cannot promote coverage of regions unsupported by the proposal. So I did not really see why this strategy encourage mode coverage. In my opinion, what proposed in this paper tends to sharpen modes rather than broaden coverage.

2. It is well known that importance sampling can exhibit high variance---an issue already present in iDEM-style neural samplers. The paper estimates the IS weights via self-normalised importance sampling (SNIS), which typically further increases estimator variance. I am particularly concerned about the variance behaviour in high-dimensional settings.

3. The evaluation is not convincing. The method is only evaluated on GMM in 2D space, along with DW and LJ. While LJ is challenging for neural samplers due to its sharp graident, its not a multi-model distribution and cannot reflect the claimed contribution of this paper. GMM is also a relative simple target (for example, what I found is that directly using TSM to draw samples can yield very good mode coverage in even 500 dims). Therefore, I would suggest the author to demonstrate the algorithm on more realistic targets.



Therefore, in summary, I am regret to say that this paper does not meet the requirement of ICLR. This paper does not address the key challenges, the experiment evaluation is not sufficient and the claim of model coverage can be misleading. The IS is known to suffer from high variance and there is no convincing evidence that this does cause issue in the proposed method in high dimensions

**Questions:**

See weakness

---

### Official Review · Reviewer_TzMe · 2025-10-25

**Soundness:** 3
**Presentation:** 2
**Contribution:** 1
**Rating:** 2
**Confidence:** 4

**Summary:**

This paper proposes adaptive, data-free diffusion training using a triple importance-weighted score matching objective. Evaluation samples are drawn via importance sampling between a replay buffer and an estimate of the marginal density, while the objective itself uses an IS estimate of the marginal score. The authors analyze the Monte Carlo error of the loss and compare performance with existing neural samplers on 2D Gaussian mixtures and classical particle potentials, including a short ablation on the role of importance sampling.

**Strengths:**

* The paper tackles an important problem

**Weaknesses:**

* The method is fundamentally unscalable with respect to dimensionality due to the reliance on importance sampling. (1) The empirical approximation of the buffer distribution does not scale (estimating the density of a dataset of samples is itself a challenging topic in probabilistic modeling) (2) It is well known that the IS-based score estimate (Eq. 7) suffers from extremely high variance when $t$ is close to $T$, and the same issue arises for the marginal density estimate (see [1], a crucial missing reference). Importantly, both the score and marginal estimates effectively perform IS on the posterior $p_{0|t} \propto p_{t|0} p_0$, which coincides with the target distribution for large $t$ (see [2], which analyzes this phenomenon). Thus, computing these estimates is as hard as sampling the target distribution itself using IS with a Gaussian proposal (an issue that likely manifested in App. I).
* The fact that the buffer’s importance weights are unbiased (but high-variance stochastic estimates) is never questioned or discussed.
* In the introduction, the authors claim that the method will enhance mode discovery for neural samplers. However, since the buffer is populated only with samples from the model itself, the procedure cannot explore regions outside those covered by the very initial buffer. Consequently, if certain modes are absent from the first buffer, they will never be discovered.
* On L194 (and in App. F), the paper claims that iDEM is simply Eq. (6) with $w = 1$, which is incorrect. Although iDEM uses an off-policy approach, information about the target density still appears in the IS estimate of the score. The correct statement is that Eq. (13) corresponds to iDEM with $w = 1$.
* The experimental evaluation is extremely limited. The synthetic experiments are restricted to 2D settings, and comparisons are made only against neural samplers, omitting established baselines such as (Adaptive) Parallel Tempering [3] and (Adaptive) Sequential Monte Carlo [4,5]. Moreover, the LRDS algorithm is excluded on the grounds that it "requires privileged access to target distribution statistics," which is incorrect (it only requires initial samples from a simple but imperfect sampler such as Langevin MCMC).

[1] RuiKang OuYang, Bo Qiang, & José Miguel Hernández-Lobato. (2025). BNEM: A Boltzmann Sampler Based on Bootstrapped Noised Energy Matching.

[2] Grenioux, L., Noble, M., Gabrié, M., & Oliviero Durmus, A. (2024). Stochastic Localization via Iterative Posterior Sampling. In Proceedings of the 41st International Conference on Machine Learning (pp. 16337–16376). PMLR.

[3] Syed, S., Bouchard-Côté, A., Deligiannidis, G., & Doucet, A. (2021). Non-Reversible Parallel Tempering: A Scalable Highly Parallel MCMC Scheme. Journal of the Royal Statistical Society Series B: Statistical Methodology, 84(2), 321-350.

[4] Saifuddin Syed, Alexandre Bouchard-Côté, Kevin Chern, & Arnaud Doucet. (2024). Optimised Annealed Sequential Monte Carlo Samplers.

[5] Yan Zhou, Adam M. Johansen, & John A.D. Aston (2016). Toward Automatic Model Comparison: An Adaptive Sequential Monte Carlo Approach. Journal of Computational and Graphical Statistics, 25(3), 701–726.

**Questions:**

* Why not use a kernel density estimation (KDE) approach for the buffer density $p_t^{\mathcal{B}}$ ? Employing Gaussian kernels could provide direct access to an estimate of the marginal density. Moreover, since the target density is known, it could be leveraged to refine this estimate.
* Is importance sampling still theoretically valid when using stochastic unbiased estimates of the importance weights?
* Have the authors considered incorporating local update steps to improve the buffer’s accuracy, as done in many adaptive training schemes (see [A], for example)?
* The nESS plots are unclear: since effective sample size should depend on time, are these plots averaged across time steps?

[A] Marylou Gabrié, Grant M. Rotskoff, & Eric Vanden-Eĳnden (2022). Adaptive Monte Carlo augmented with normalizing flows. Proceedings of the National Academy of Sciences, 119(10), e2109420119.

---

### Official Review · Reviewer_n6jF · 2025-10-27

**Soundness:** 3
**Presentation:** 2
**Contribution:** 2
**Rating:** 2
**Confidence:** 4

**Summary:**

The paper proposes a new training method for a score based diffusion model in the context of sampling from a distribution known through its un-normalized probability density function and therefore without access to training samples. The paper advocates for maximizing the forward KL divergence which is estimated in practice using a series of importance sampling (IS) approximation. The expectation in the definition of the KL uses a first IS estimation in the same manner as (Muller et al 2019) did with normalizing flows. The weights of this first IS estimation being intractable themselves, they are estimated with IS approximations (numerator and denominator) that are of the same nature as the last IS approximation of the approach used to estimate the target score. This one is  inspired from Akhound-Sageh et al 2024 and Bortoli et al 2024.

Analysis of the scaling with respect to sampling size of the IS approximations are provided. Numerical experiments report competitive result in usual but limited benchmarks.

**Strengths:**

1. The topic of samplers assisted with generative models is timely.
2. The theoretical analysis extends to bounds on the variance which is a known limitation of IS (but does not include the scaling in dimension).

**Weaknesses:**

3. The method is a rather limited modification of the existing approach iDEM. The numerical tests while showing marginal improvements in terms of absolute value on the reported metrics, these are very probably marginal in terms of the accuracy in estimating physical observables (see next point).
4. The numerical evidence for the proposed approach over existing ones is limited.
	- The approach relies on importance sampling which is expected to suffer from a curse of dimensionality but no systematic investigation is conducted for increasing dimension. See for instance the proposition of [Grenioux2025]
	-  The marginal improvements reported over iDEM are not necessarily meaningful as far as the sampling task is concerned. Wasserstein and TV distances are difficult to interpret and not physically relevant. It would be much more interesting to report the ability to find modes and compute relative weights (aka free energy differences in the language of physics). For instance for LJ13 and LJ55, Figure 5 in the appendix (showing histograms of energies and pairwise distances) reveals that the proposed methods improves really marginally over iDEM and that neither iDEM nor the proposed approach offers a physically convincing sampling of the target distribution.
5. The proposed theoretical analysis focuses on scaling with the sampling size of the IS estimator and report usual Monte Carlo scalings. As such it misses the point that the pre-factor might scale very poorly with dimension, as it is usually the case with importance sampling, and be the major bottleneck to the accuracy of the estimation rather than the number of sample L.
6. The related literature is discussed too rapidly in the main text and sometimes in a misleading fashion:
	- Although it is impossible to cite exhaustively works using generative models for sampling the authors should consider citing pioneering approaches [Wu2019] and Gabrié et al 2022 (already cited in appendix).
	- On line 48, I am a bit confused by the authors’ choice of only citing He et al 2024 for a data free setting as it is the case of all the previously given references, maybe I misunderstood the point.
	- About the shortcomings of the reverse KL training, the introduction is currently very misleading by stating that “Previous methods ignore this mismatch”, and leaving the reader with the impression that the paper is among the first to notice the relevance of the forward KL when training a probabilistic model for sampling:
		- Müller et al 2019 are not cited in the main text while they are proposing the pendant of the article for normalizing flows (NFs).
		- This is why Noé et al 2019 use some short MD trajectories as well as their training by energy.
		- It motivated combining learning and annealing for better mode coverage as soon as 2019,2020 [Wu2019, McNaughton2020] and since reused by [Hackett2021] and [Ciarella2023] for instance.
		- It also motivated using adaptive methods that are now known as “replay buffer methods”  first proposed for generative models by Gabrié et al 2022 for NF,  training samples being here generated by a Metropolis-Hastings MCMC using proposals from the NF being trained. Without deep learning but using simpler parametric models, this idea of using the training models to estimate a forward KL objective was even investigated previously [Parno2018,Naesseth2020].  Although some of these works are cited in the related works in the appendix, they seem to belong to line 078-081 in the introduction.


Minor:

7.  Writing: The authors do not state clearly in their introduction what they intend to do, for instance the fact that they are focusing on diffusion models.
8. Line 165 - “Unknown score function $\nabla \log p_t(x_t)$: The target of the inner squared error, the true score $\nabla log p_t(x_t)$, is itself unknown because $p_t$ depends on $\pi$ with unknown normalization constant Z” - This is not the reason why the score is not known, the partition function does not intervene in the score, however, computing the score at a non-zero noise level requires to perform a high-dimensional integral which is intractable which yields one of the IS approximation of the proposed approach.

Additional refs:

- [Wu2019] Wu, Dian, Lei Wang, and Pan Zhang. “Solving Statistical Mechanics Using Variational Autoregressive Networks.” Physical Review Letters 122, no. 8 (2019): 080602. https://doi.org/10.1103/PhysRevLett.122.080602.
[McNaughton2020] McNaughton, B., M. V. Milošević, A. Perali, and S. Pilati. “Boosting Monte Carlo Simulations of Spin Glasses Using Autoregressive Neural Networks.” Physical Review E 101, no. 5 (2020): 053312. https://doi.org/10.1103/PhysRevE.101.053312.
- [Hackett 2021] Hackett, Daniel C., Chung-Chun Hsieh, Michael S. Albergo, et al. “Flow-Based Sampling for Multimodal Distributions in Lattice Field Theory.” arXiv:2107.00734. Preprint, arXiv, July 1, 2021. http://arxiv.org/abs/2107.00734.
- [Ciarella2023] Ciarella, Simone, Jeanne Trinquier, Martin Weigt, and Francesco Zamponi. “Machine-Learning-Assisted Monte Carlo Fails at Sampling Computationally Hard Problems.” Machine Learning: Science and Technology 4, no. 1 (2023): 010501. https://doi.org/10.1088/2632-2153/acbe91.
- [Naesseth2020] Naesseth, Christian, Fredrik Lindsten, and David Blei. “Markovian Score Climbing: Variational Inference with KL(P\vert \vert q).” Advances in Neural Information Processing Systems 33 (2020): 15499–510. https://proceedings.neurips.cc/paper_files/paper/2020/hash/b20706935de35bbe643733f856d9e5d6-Abstract.html.
- [Parno2018] Parno, Matthew D., and Youssef M. Marzouk. “Transport Map Accelerated Markov Chain Monte Carlo.” SIAM/ASA Journal on Uncertainty Quantification 6, no. 2 (2018): 645–82. https://doi.org/10.1137/17M1134640.
- [Grenioux2025] Grenioux, Louis, Maxence Noble, and Marylou Gabrié. “Improving the Evaluation of Samplers on Multi-Modal Targets.” Paper presented at Frontiers in Probabilistic Inference: Learning meets Sampling. ICLR Workshop on Frontiers in Probabilistic Inference: Learning Meets Sampling, April 24, 2025. https://openreview.net/forum?id=d91E9RhVFU.

**Questions:**

9. How were the hyper parameters of each of the compared method chosen? Is the score architecture the same for all methods?
10. How important is the addition of uniform/flat gaussian sample to the method? Does it mainly explain the advantage found over iDEM (namely, what is the performance of iDEM if using these as well in its replay buffer)? These uniform/flat gaussian distributions are a priori unlikely to be effective in high dimension.
11. Reporting the ESS in figure 2 is interesting. Why is it not given in Appendix G.6 for LJ55?  A systematic picture as a the dimensionality increases, for example of mixtures of Gaussians to be simple (see again [Grenioux2025], would be particularly relevant. )

---

### Official Review · Reviewer_eJQe · 2025-10-28

**Soundness:** 2
**Presentation:** 3
**Contribution:** 2
**Rating:** 2
**Confidence:** 5

**Summary:**

The paper extends the

iDEM objective:
$$L_{\text{IDEM}}(\theta) = \iint || s_{\theta}(x_t, t) - \nabla_{x_t} \log p_t(x_t) ||^2 p_t^{B} (x_t) dx_t \ dt$$
to
$$L_{new}(\theta) = \iint || s_{\theta}(x_t, t) - \nabla_{x_t} \log p_t(x_t) ||^2  \frac{p_t(x_t)}{p^{B}_t (x_t)} p^{B}_t (x_t) dx_t \ dt$$

where the ratio $\frac{p_t(x_t)}{p^{B}_t (x_t)} $ is estimated by a importance sampling -based estimator.

**Strengths:**

The writing is OK, there are many theorems which make it look good. The paper is long and covers many details.

**Weaknesses:**

**Fundamental Error:**

The iDEM objective $$L_{\text{IDEM}}(\theta) = \iint || s_{\theta}(x_t, t) - \nabla_{x_t} \log p_t(x_t) ||^2 p_t^{B} (x_t) dx_t \ dt$$ is not the fisher divergence between $p_t^B$ and $p_t^\theta$ (line 1326 in the paper), since the score inside the L2 norm is $p_t$. The Fisher divergence between $p_t^B$ and $p_t^\theta$  is  $$FD = \iint || s_{\theta}(x_t, t) - \nabla_{x_t} \log p^B_t(x_t) ||^2 p_t^{B} (x_t) dx_t \ dt.$$

Why is this important? Since in the IDEM, when the support $p_t^B$ covers the support of $p_t$ and $p^\theta_t$, it actually defines a valid divergence between $p_t$ and $p^\theta$, it won't give any bias in the distribution learning. Intuitively, the score of two distributions matches everywhere in their support means two distributions are the same.

**Claim is invalid:**

This is not a small mistake that appears in the appendix, but it makes the whole claim invalid. The goal of the paper, including title, is to "ENHANCED MODE COVER-AGE". But if you compare iDEM and the proposed new objective
$$L_{new}(\theta) = \iint || s_{\theta}(x_t, t) - \nabla_{x_t} \log p_t(x_t) ||^2  \frac{p_t(x_t)}{p^{B}_t (x_t)} p^{B}_t (x_t) dx_t \ dt$$

The difference is the ratio; we all know the ratio is only effective when $p^B_t$ and $p_t$ share the same support (with non-negligible density). Therefore, in this process, all the mode covering ability comes from $p_B$, the additional ratio won't give it any mode covering ability, and another importance sampling-estimation of the ratio will just increase its bias in principle.

Therefore, the whole claim of the paper does not hold. It may be possible that the additional ratio will make it converge faster due to the gradient being more efficient or unbiased if the goal is to minimize the **Fisher Divergence**, but you then need to argue why Fisher divergence is better than the divergence defined by iDEM. It also requires rewriting the paper's claim and conducting different experiments.

A minor thing is that importance sampling-based estimation may not be effective in high dimensions, but it is a minor point since non-convex global optimisation in high dimensions is generally impossible.

**Questions:**

See above

---

### Note · Authors · 2025-12-01

I have read and agree with the venue's withdrawal policy on behalf of myself and my co-authors.